# AGUVIS: UNIFIED PURE VISION AGENTS FOR AUTONOMOUS GUI INTERACTION

## ABSTRACT

Graphical User Interfaces (GUIs) are critical to human-computer interaction, yet automating GUI tasks remains challenging due to the complexity and variability of visual environments. Existing approaches often rely on textual representations of GUIs, which introduce limitations in generalization, efficiency, and scalability. In this paper, we introduce AGUVIS, a unified pure vision-based framework for autonomous GUI agents that operates across various platforms. Our approach leverages image-based observations, and grounding instructions in natural language to visual elements, and employs a consistent action space to ensure cross-platform generalization. To address the limitations of previous work, we integrate explicit planning and reasoning within the model, enhancing its ability to autonomously navigate and interact with complex digital environments. We construct a large-scale dataset of GUI agent trajectories, incorporating multimodal reasoning and grounding, and employ a two-stage training pipeline that first focuses on general GUI grounding, followed by planning and reasoning. Through comprehensive experiments, we demonstrate that AGUVIS surpasses previous state-of-the-art methods in both offline and real-world online scenarios, achieving, to our knowledge, the first fully autonomous pure vision GUI agent capable of performing tasks independently without collaboration with external closed-source models. We will open-source all datasets, models, and training recipes to facilitate future research.

## 1 INTRODUCTION

Graphical User Interfaces (GUIs) are a cornerstone of human-computer interaction, providing a structured yet intuitive platform for users to accomplish tasks across various digital environments: website, desktop, and mobile devices (Deng et al., 2023; Zhou et al., 2024; Xie et al., 2024; Rawles et al., 2024b). Automating GUI operations through autonomous agents can revolutionize productivity by enabling seamless task execution on various applications using existing human-centric tools. Moreover, this approach lays the groundwork for advanced AI systems that can interact with and learn from rich digital environments in ways that mirror human behavior.

To effectively perform GUI tasks autonomously, a GUI agent requires three core competencies: understanding, grounding, and planning & reasoning. For GUI understanding, the agent must first comprehend high-resolution and complex interfaces designed for human users, enabling it to grasp the context and perform subsequent reasoning tasks. GUI grounding involves mapping natural language instructions to visual observations of the interface. For planning and reasoning, the agent must synthesize and analyze the current multimodal observations of the environment with previous observations and action histories, enabling it to generate coherent and effective next steps to ultimately achieve the task goal. Although recent advances in large vision-language models (LVLMs) (OpenAI, 2024; Reid et al., 2024; Li et al., 2024a; Wang et al., 2024a) have significantly enhanced the ability of AI systems to interpret complex visual interfaces, there remain critical challenges in grounding and reasoning specifically tailored for GUI tasks. We identify three primary challenges that must be addressed to advance the capabilities of GUI agents:

**Enhancing Pure Vision Framework.** Previous approaches (Gur et al., 2024; Kim et al., 2023; Deng et al., 2023; Zhou et al., 2024; Xie et al., 2024) predominantly focus on mapping natural language instructions to textual representations of GUIs, such as HTML or accessibility trees. This method presents several limitations. Firstly, GUIs are inherently visual, and leveraging image-based repre-

sentations aligns more closely with human cognitive processes. Secondly, textual representations can vary widely across different environments, complicating the generalization of the model and limiting the availability of consistent training data. Finally, these textual representations are often verbose and complex, leading to increased inference times compared to more compact image encodings (Figure 2). By unifying observations across platforms as images and grounding instructions to image coordinates, GUI agents can generalize more effectively across diverse environments.

**Unification Across GUI Environments.** The action spaces and control APIs for GUI interactions vary significantly across diverse environments, particularly when the observations are textual. Even within the same platform, the action space can differ greatly. This heterogeneity limits the amount of training data available for each environment, impeding the development of a model that can generalize effectively across different platforms and scale further. A unified action space that abstracts these environmental differences is crucial for creating robust and adaptable GUI agents. Previous work (Chen et al., 2024b; Zeng et al., 2024) has attempted to unify digital agent data across diverse environments, such as combining GUI, game, and CLI interfaces for joint training. However, these interfaces do not share the same interaction logic. In contrast, GUIs on desktop, web, and mobile platforms naturally share similar human-computer interaction (HCI) logic. This commonality facilitates their unification, enabling consistent visual observations and action spaces that mutually benefit both visual grounding and reasoning.

**Integrating Planning and Reasoning with Grounding.** Current methodologies (Zheng et al., 2024a) often depend on the reasoning capabilities of closed-source large language models (LLMs) (OpenAI, 2024) to plan the completion of GUI tasks or, alternatively, train agents to make direct action decisions through grounding without an explicit reasoning process. This dichotomy results in either a lack of grounding abilities or a lack of comprehensive reasoning abilities. Recently, some works (Gou et al., 2024; Lu et al., 2024) attempt to use closed-source LLMs with specialized GUI grounding models together and communicate with natural language instruction to utilize both abilities. However, on the one hand, natural language communication between the two models usually results in information loss. On the other hand, most importantly, this approach is not further scalable to solve GUI interaction since grounding has been improved close to the upper bound with data synthesis, and most remaining problems are planning related. However, the GUI planning and reasoning ability of closed-source LLMs cannot be further improved.

To address these challenges, we introduce a unified framework for GUI agents that harmonizes pure vision observation and consistent action spaces across diverse environments. Our approach leverages vision-based grounding to improve generalization and reduce inference costs while employing a standardized action space with a plugin system to facilitate consistent learning and interaction across various platforms. After a unified GUI grounding training stage, we demonstrate that unified augmented datasets can effectively build a model capable of executing complex GUI grounding instructions on various platforms. In addition, we integrate explicit visual planning and reasoning into the same model, enabling autonomous navigation and interaction within complex digital environments. Since existing GUI agent trajectories do not fully support these demands, we have unified the existing planning datasets on different platforms and constructed a large-scale, pure vision, cross-platform, multi-step dataset of agent trajectories, featuring comprehensive multimodal reasoning and grounding. Through extensive experiments across various scenarios, we demonstrate the effectiveness of our approach in advancing the state-of-the-art for pure vision-based autonomous GUI agents. To our knowledge, this is the first model that can autonomously complete tasks in real-world online environments without relying on higher reasoning abilities from closed-source models.

Our contributions are as follows:

- We introduce a unified pure vision framework for building generalizable GUI agents that operate with vision-based observations and a plugin-enabled action system, enhancing cross-platform adaptability.

- We develop a comprehensive data pipeline that unifies existing GUI grounding annotations and integrates explicit planning and reasoning. This enables the construction of large-scale datasets for grounding and multi-step agent trajectory datasets across platforms.

- Starting with a VLM, we present a two-stage training process—first for GUI grounding, followed by planning and reasoning—resulting in AGUVIS, the first cross-platform au-

tonomous GUI agent capable of performing complex tasks independently without relying on closed-source models. All data, models, and training resources will be open-sourced.

# 2 AGUVIS

## 2.1 PROBLEM FORMULATION

We model the autonomous GUI agent's interaction with the environment as a Partially Observable Markov Decision Process (POMDP), characterized by the tuple $(\mathcal{S}, \mathcal{A}, \mathcal{O}, T, O)$. In this formulation, $\mathcal{S}$ represents the set of possible states of the environment, $\mathcal{A}$ denotes the set of actions the agent can take, and $\mathcal{O}$ refers to the set of observations the agent can receive. The state transition function, $T : \mathcal{S} \times \mathcal{A} \times \mathcal{S} \to [0, 1]$, defines the probability of transitioning from one state to another given an action, while the observation function, $O : \mathcal{S} \times \mathcal{A} \times \mathcal{O} \to [0, 1]$, specifies the probability of receiving a particular observation given a state and an action.

At each time step $t$, the agent receives an image observation $o_t$ from the GUI environment, reasons and generates an inner monologue (Huang et al., 2022) based on its previous actions and observations. This inner monologue consists of three components: a natural language description of the current observation ($d_t$), internal reasoning ($h_t$) based on the high-level goal $G$, the observation description $d_t$, and previous thoughts $h_{t-1}$, and finally, a low-level action instruction ($a_t^{\text{instr}}$) in natural language that specifies the next action. The agent then executes the action $a_t$ based on the instruction $a_t^{\text{instr}}$, receives a new observation $o_{t+1}$, and repeats this process until it either achieves the goal $G$ or reaches a terminal state.

## 2.2 UNIFIED PURE VISION FRAMEWORK

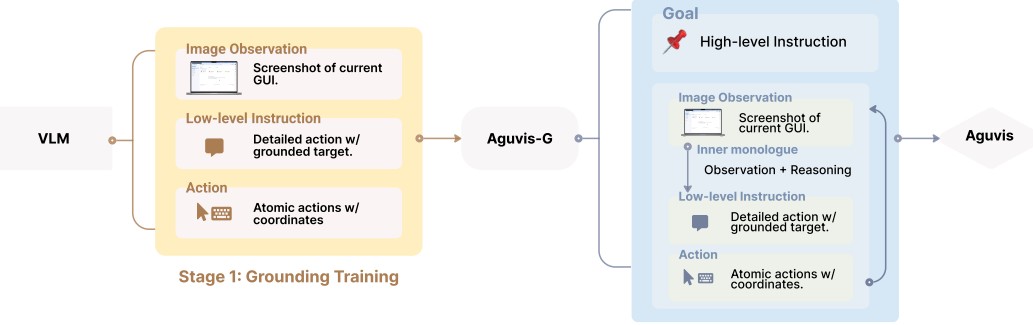

Figure 1: Overview of the two-stage training paradigm for autonomous GUI agents.

In this work, we propose to unify observation and action space via pure vision and `pyautogui` commands with a pluggable action system (Table 9). For observation, pure vision does not require the model to understand different UI source codes of the interfaces of different platforms, such as HTML of the webpage, and accessibility tree of desktop and mobile operating systems, which can help improve the generalization. Meanwhile, pure vision can reduce the input token length. Generally, the input length of accessibility tree observation is 6k tokens (Xie et al., 2024), and HTML is 4k tokens (Figure 2), depending on the complexity of the interface. Compared with relatively long input, the token cost of image observation does not vary across different interfaces but only depends on model design, which in our case is 1200 tokens for 720p image observation.

For unified action space, we choose the widely used standard `pyautogui` action space with a pluggable action system. This library leverages the high-level programming language Python to replicate and replay various human inputs into computers through code, allowing us to construct a universal and complete representation of actions. We show the action space in Table 9. We use `pyautogui` commands to unify basic GUI operations of all platforms including web, desktop, and mobile. Over this action space, an agent model can then learn to generate actions in order to control GUI without any action space description.

While mouse and keyboard inputs form the core of GUI interactions, they are not comprehensive. Certain platforms require additional actions. For example: (1) specific actions on mobile platforms such as swiping; (2) shortcuts that efficiently perform a series of actions like opening apps; (3) communication actions such as providing answers or terminating after completion. To address these extended requirements, we introduce a pluggable action system. This system allows us to expand the action space by aligning new actions with the existing `pyautogui` commands where possible. For actions that cannot be directly mapped, the pluggable system provides the flexibility to incorporate them with detailed action descriptions. This enables the model to generalize effectively to environments where new actions are introduced. By combining pure vision observations with a unified action space and a flexible pluggable system, our framework enables the training of a single model that can operate across diverse platforms. This setup not only simplifies the training process but also ensures the model can generalize and adapt to novel environments and tasks.

## 2.3 THE AGUVIS COLLECTION

GUI agent trajectories are a low-resource data source compared with its challenges. This is because the observation and action space vary across different environments even on the same platform. Fortunately, GUI environments share the same operation logic and similar action space. We can efficiently unify existing data to scale the training set. Therefore, we propose THE AGUVIS COLLECTION, a large-scale GUI agent training dataset collected and augmented with existing GUI agent data. This data collection consists of two splits: grounding split (Table 10) and planning & reasoning split (Table 11), corresponding to the two important GUI abilities.

**Template-augmented Grounding Data.** Vision-based grounding requires the model to ground the natural language intent to the image observation with coordinates. On one hand, there are several previous works that have built datasets on different platforms, including natural language instructions and corresponding target elements. We collected and unified them into `pyautogui` commands format. On the other hand, we found that there are many datasets proposed for user interfaces on different platforms that contain a large amount of metadata, including the positions of all text/icons/widgets in the current interface. Using this type of data we constructed templates for `pyautogui` actions. We randomly generated grounding data pairs through these templates to train models to ground these elements based on images. This operation greatly expanded the data scale.

**VLM-augmented Planning & Reasoning Trajectories.** High-quality GUI agent trajectories contain several key components: a high-level goal, a sequence of interleaved observations, natural language reasoning, and grounded actions. Existing approaches typically rely on human annotation to collect these trajectories (Deng et al., 2023; Rawles et al., 2024b; Li et al., 2024c). Most of the agent trajectory data contains high-level goals, observations, and grounded actions. However, the intermediate reasoning process and low-level action instructions are not included. This makes it difficult for existing data to train agents to perform chain-of-thought or inner monologue reasoning to help the model plan the next action, resulting in poor agent performance.

To augment the agent trajectories with detailed reasoning and low-level action instructions, we employ a vision-language model (VLM) to generate the inner monologue for each step in the trajectory. Specifically, for each time step $t$, given the high-level goal $G$, the current image observation $o_t$, and the grounded action $a_t$, we prompt the VLM to produce the inner monologue components: observation description $d_t$, thoughts $h_t$, and low-level action instruction $a_t^{\text{instr}}$. To assist the VLM in generating accurate and contextually relevant monologues, we highlight the target element associated with the grounded action $a_t$ on the image observation $o_t$. This visual cue helps the model focus on the relevant part of the interface. Additionally, we include the previous low-level action instructions $a_1^{\text{instr}}, a_2^{\text{instr}}, \ldots, a_{t-1}^{\text{instr}}$ to provide the VLM with the action history, ensuring continuity and coherence in the generated reasoning.

The prompting strategy is carefully crafted to guide the VLM in generating inner monologues that are predictive and goal-oriented, without relying on hindsight or revealing future actions. By simulating the agent's thought process in a first-person perspective, we encourage the generation of actionable instructions that align with the high-level goal and current observation. This approach results in a large-scale dataset of agent trajectories enriched with detailed reasoning and instructions.

## 2.4 Model Architecture

Unlike grounding agents that rely on structured UI representations (such as accessibility trees) as their textual input, vision-based grounding requires the model to map intents directly to visual observations. This means the model needs to encode high-resolution images while preserving their original aspect ratios. Recent advances in VLMs have made these capabilities possible. We choose Qwen2-VL (Wang et al., 2024b) as our starting VLM. It uses NaViT as an image encoder with native dynamic resolution support (Dehghani et al., 2023). Unlike its predecessor, Qwen2-VL can now process images of any resolution, dynamically converting them into a variable number of visual tokens. To support this feature, ViT is modified by removing the original absolute position embeddings and introducing 2D-RoPE (Su et al., 2024) to capture the two-dimensional positional information of images. Based on these unique features, Qwen2-VL is highly suitable for GUI agents' needs. It can encode high-resolution images of any ratio with relatively fewer image token costs. Therefore, we chose Qwen2-VL as our starting VLM to build our GUI agent.

LLaVA-OneVision (Li et al., 2024a) is another suitable VLM as it also supports high-resolution any ratio image encoding, although its image token cost is relatively higher than Qwen2-VL. We also apply our data recipe and training strategy to LLaVA and show that our framework is model-independent and generally works for high-resolution VLMs details are shown in Section 4.2..

## 2.5 Training Paradigm

We begin with a Vision-Language Model (VLM) that possesses advanced image understanding capabilities, and the training process is divided into two main stages: Grounding Training and Planning & Reasoning Training. Each stage utilizes a distinct data split from our THE AGUVIS COLLECTION to progressively enhance the VLM's abilities.

**Stage 1: Grounding Training** In this stage, we focus on enabling the model to understand and interact with objects within a single GUI screenshot. GUI environments typically feature multiple interactable objects within a single screenshot, generating a large volume of grounding data but leading to shorter, less diverse interaction sequences, which can limit training efficiency.

We train our model with a grounding packing strategy where multiple instruction-action pairs are bundled into a single image, resulting in a single-image-multiple-turn format. This technique allows the model to process several grounding examples from one screenshot, reducing redundant training overhead while retaining a high level of grounding performance. This approach significantly accelerates training by maximizing the use of each image without compromising accuracy. To equip our model with the capability for GUI understanding and grounding, which serves as the foundation for subsequent planning and reasoning, we conducted this training stage. Upon completing Stage 1 training, the model is referred to as AGUVIS-G.

**Stage 2: Planning & Reasoning Training** Building on the foundation of AGUVIS-G, the second stage introduces more complex decision-making and reasoning processes. This phase is designed to teach the model how to execute multi-step tasks by reasoning through agent trajectories that vary in complexity and environments, encompassing diverse reasoning modes.

Thanks to our detailed inner monologue trajectory data, we implement a reasoning mixture approach, where the model is exposed to various levels of cognitive complexity, from straightforward low-level action instructions to full inner monologues that include observation descriptions, thoughts, and detailed action plans. By dynamically adjusting the complexity of these trajectories, we train the model to be adaptable, fostering step-by-step reasoning and high-level decision-making abilities. This diversity in reasoning ensures that the model can handle a wide range of tasks with nuanced understanding and precision. After this stage, the fully trained model is called AGUVIS, which can be employed in both offline and online GUI tasks across diverse environments.

## 3 Experiments

To evaluate the effectiveness of GUI agent models on various platforms, we conduct experiments on several GUI benchmarks: GUI Grounding Evaluation and Offline/Online GUI Agent Evaluation.

## 3.1 GUI GROUNDING EVALUATION

Table 1: Comparison of various planners and grounding methods on ScreenSpot across various device and input modalities. The top part of table shows the results on *original instructions* evaluation setting while the bottom part shows results on *self-plan* evaluation setting. Best results are in bold.

| Planner | Grounder | Mobile | | Desktop | | Web | | Avg |
|---------|----------|--------|-------------|---------|-------------|------|-------------|-----|
| | | Text | Icon/Widget | Text | Icon/Widget | Text | Icon/Widget | |
| - | GPT-4 | 22.6 | 24.5 | 20.2 | 11.8 | 9.2 | 8.8 | 16.2 |
| | GPT-4o | 20.2 | 24.9 | 21.1 | 23.6 | 12.2 | 7.8 | 18.3 |
| | CogAgent | 67.0 | 24.0 | 74.2 | 20.0 | 70.4 | 28.6 | 47.4 |
| | SeeClick | 78.0 | 52.0 | 72.2 | 30.0 | 55.7 | 32.5 | 53.4 |
| | Qwen2-VL | 75.5 | 60.7 | 76.3 | 54.3 | 35.2 | 25.7 | 55.3 |
| | UGround | 82.8 | 60.3 | 82.5 | 63.6 | 80.4 | 70.4 | 73.3 |
| | AGUVIS-G-7B | **88.3** | **78.2** | **88.1** | **70.7** | **85.7** | **74.8** | **81.8** |
| GPT-4 | SeeClick | 76.6 | 55.5 | 68.0 | 28.6 | 40.9 | 23.3 | 48.8 |
| | OmniParser | 93.9 | 57.0 | 91.3 | 63.6 | 81.3 | 51.0 | 73.0 |
| | UGround | 90.1 | 70.3 | 87.1 | 55.7 | 85.7 | 64.6 | 75.6 |
| GPT-4o | SeeClick | 81.0 | 59.8 | 69.6 | 33.6 | 43.9 | 26.2 | 52.3 |
| | UGround | 93.4 | 76.9 | 92.8 | 67.9 | 88.7 | 68.9 | 81.4 |
| | AGUVIS-7B | **95.6** | 77.7 | 93.8 | 67.1 | 88.3 | 75.2 | **84.4** |
| | AGUVIS-72B | 94.5 | **85.2** | 95.4 | 77.9 | 91.3 | 85.9 | 89.2 |

**ScreenSpot.** We first assess the performance of GUI grounding, which is a foundational capability of GUI agent models. Following previous work (Cheng et al., 2024; Gou et al., 2024), we evaluate models on ScreenSpot (Cheng et al., 2024). This dataset encompasses a variety of grounding instructions tailored for mobile, desktop, and website platforms, and is assessed under two distinct settings: (1) *Original Instructions*: models perform grounding actions directly following the original instructions; and (2) *Self-plan*: models are required to generate plans in natural language based on the original instructions before executing grounding actions.

The performance illustrated in Table 1 demonstrates that AGUVIS exhibits impressive GUI grounding capabilities under two settings across various platforms. We observe that with the proposed grounding training, AGUVIS-G-7B significantly outperforms existing models with the original instructions, suggesting that AGUVIS has strong universal GUI grounding capability. After training on high-quality planning trajectory data, AGUVIS shows strong planning capability and outperforms previous models that rely on external closed-source LLMs (like GPT-4o). Moreover, further scaling parameters, AGUVIS-72B achieves state-of-the-art performance, attaining an average score of 89.2.

Table 2: Performance comparison on Multimodal Mind2Web across different settings. We report element accuracy (Ele.Acc), Operation F1 (Op.F1), and step success rate (Step SR). Best results are in bold. "T" means the textual HTML code as inputs. "I" means the GUI images as inputs. More explanation about result source in Appendix D.2

| Obs. | Planner | Grounder | Cross-Task | | | Cross-Website | | | Cross-Domain | | |
|------|---------|----------|------------|-------|---------|------------|-------|---------|------------|-------|---------|
| | | | Ele.Acc | Op.F1 | Step SR | Ele.Acc | Op.F1 | Step SR | Ele.Acc | Op.F1 | Step SR |
| T | GPT-3.5 | Choice | 19.4 | 59.2 | 16.8 | 14.9 | 56.5 | 14.1 | 25.2 | 57.9 | 24.1 |
| | GPT-4 | Choice | 40.8 | 63.1 | 32.3 | 30.2 | 61.0 | 27.0 | 35.4 | 61.9 | 29.7 |
| T + I | GPT-4 | Choice | 46.4 | 73.4 | 40.2 | 38.0 | 67.8 | 32.4 | 42.4 | 69.3 | 36.8 |
| | GPT-4 | SoM | 29.6 | - | 20.3 | 20.1 | - | 13.9 | 27.0 | - | 23.7 |
| I | - | SeeClick | 23.8 | - | - | 15.3 | - | - | 16.2 | - | - |
| | - | CogAgent | 54.2 | - | - | 50.0 | - | - | 54.7 | - | - |
| I | GPT-4o | SeeClick | 32.1 | - | - | 33.1 | - | - | 33.5 | - | - |
| | GPT-4V | OmniParser | 42.4 | 87.6 | 39.4 | 41.0 | 84.8 | 36.5 | 45.5 | 85.7 | 42.0 |
| | GPT-4o | UGround | 47.7 | - | - | 46.0 | - | - | 46.6 | - | - |
| I | AGUVIS-7B | | 64.2 | 89.8 | 60.4 | 60.7 | 88.1 | 54.6 | 60.4 | **89.2** | 56.6 |
| | AGUVIS-72B | | **69.5** | **90.8** | **64.0** | **62.6** | **88.6** | **56.5** | **63.5** | 88.5 | **58.2** |

## 3.2 OFFLINE GUI AGENT EVALUATION

**Multimodal-Mind2Web.** We utilize Multimodal-Mind2Web (Zheng et al., 2024a) for evaluating the offline planning capabilities of GUI agents on websites, which builds on the original Mind2Web (Deng et al., 2023). We compare with previous work including closed LLMs taking text-only (Deng et al., 2023) or SoM as inputs (Zheng et al., 2024a) and recent prue vision-based agent models. Following previous work (Cheng et al., 2024; Gou et al., 2024), AGUVIS only use the GUI screenshot as observation. We report element accuracy (Ele.Acc), Operation F1 (Op.F1), and step success rate (Step SR). As shown in Table 2, AGUVIS consistently achieves superior performance, with a notable improvement in Step SR ($+51.9\%$ averaged), indicating enhanced reasoning capabilities regarding planning.

**AndroidControl.** We assess the planning performance of GUI agent models on mobile devices using AndroidControl (Li et al., 2024d). Following the setting in Li et al. (2024d), we randomly sample 500 step-actions to create a subset, and we report the step accuracy on out-of-domain (OOD) data within both high-level and low-level tasks. The high-level task setting necessitates that the model plans and executes actions, whereas the low-level task setting requires the model to simply adhere to human-labeled instructions for executing the next-step action. We compare with baselines that take textual accessibility tree or images as GUI observations. Table 3 shows that AGUVIS achieves the best performance under both settings.

Table 3: Step Accuracy of out-of-domain (OOD) data on AndroidControl under high-level tasks and low-level tasks. Best performance is in bold. "Acc.Tree" means the textual accessibility tree.

| Observation | Planner | Grounder | Step Accuracy | |
| --- | --- | --- | --- | --- |
| | | | **High-Level** | **Low-Level** |
| Acc. Tree | GPT-4-Turbo | Choice | 42.1 | 55.0 |
| | PaLM 2S (Specialized) | Choice | 58.5 | 77.5 |
| Image | GPT-4-Turbo | SeeClick | 39.4 | 47.2 |
| | GPT-4-Turbo | UGround | 46.2 | 58.0 |
| | GPT-4o | SeeClick | 41.8 | 52.8 |
| | GPT-4o | UGround | 48.4 | 62.4 |
| Image | AGUVIS-7B | | **61.5** | **80.5** |
| | AGUVIS-72B | | **66.4** | **84.4** |

## 3.3 ONLINE GUI AGENT EVALUATION

Beyond offline planning, we test AGUVIS on real-time interaction benchmarks: Mind2Web-Live (Pan et al., 2024b), AndroidWorld (Rawles et al., 2024a) and MobileMiniWob (Rawles et al., 2024b). We introduce each benchmark below and more details are shown in D.3

**Mind2Web-Live.** Mind2Web-Live is a dynamic dataset in a real web-based environment derived from the original Mind2Web. The benchmark evaluates whether each required step within a task has been completed and uses the task success rate (Task SR) as the reported metric.

**AndroidWorld.** AndroidWorld is a benchmark operating on an Android virtual environment, capable of dynamically instantiating with randomly generated parameters to generate unique tasks for automatic evaluation. To assess the pure vision agent models, we follow the instructions in Rawles et al. (2024b), installing a Pixel 6 phone simulator on our computers to serve as the experimental environment. The AndroidWorld benchmark incorporates a fully automated task-level evaluation system that automatically assesses whether a state has successfully completed a designated task.

**MobileMiniWob.** MobileMiniWob is the instantiation of 92 tasks from MiniWob++ (Zheng et al., 2024b) in AndroidWorld environment. Thus, we adopt the same observation and action space utilized in AndroidWorld and use a real-time evaluation function to determine task success rate.

Table 4: Task Success Rate (SR) and efficiency costs on Mind2Web-Live. USD Efficiency is calculated by dividing the model's total inference cost in USD by the number of successful steps.

| Inputs | Planner | Grounder | Task SR | USD Efficiency |
|--------|---------|----------|---------|----------------|
| HTML | GPT-4-Turbo | Choice | 21.1 | - |
| | GPT-4o | Choice | 22.1 | 0.142 |
| | Llama-3.1-405B | Choice | 24.0 | 0.174 |
| | Llama-3.1-70B | Choice | 20.2 | 0.031 |
| | GPT-3.5-turbo | Choice | 17.3 | 0.092 |
| Image | GPT-4-Turbo | UGround | 23.1 | - |
| | GPT-4o | UGround | 19.2 | - |
| | GPT-4o | AGUVIS-7B | **24.0** | **0.106** |
| Image | AGUVIS-72B | | **27.1** | **0.012** |

Figure 2: Comparison of Input Tokens per Step and USD Efficiency in GUI Interaction. The bar chart shows the input tokens required per step during GUI interactions, while the line graph illustrates USD Efficiency for all models.

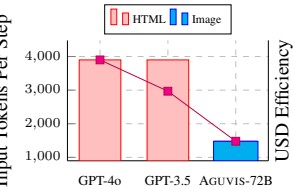

Table 5: Task Success Rates (SR) on AndroidWorld and MobileMiniWob. Best results are in bold.

| Input | Planner | Grounding | AndroidWorld$_{SR}$ | MobileMiniWob$_{SR}$ |
|-------|---------|-----------|---------------------|----------------------|
| AXTree | GPT-4-Turbo | Choice | 30.6 | 59.7 |
| | Gemini 1.5 Pro | Choice | 19.4 | 57.4 |
| Image + AXTree | GPT-4-Turbo | SoM | 25.4 | 67.7 |
| | Gemini 1.5 Pro | SoM | 22.8 | 40.3 |
| Image | GPT-4-Turbo | UGround | 31.0 | - |
| | GPT-4o | UGround | 32.8 | - |
| | GPT-4o | AGUVIS-7B | **37.1** | **55.0** |
| Image | AGUVIS-72B | | **26.1** | **66.0** |

Table 6: Success rate on the OSWorld benchmark in a screenshot-only setting

| Planner | Grounding | Task SR |
|---------|-----------|---------|
| GPT-4o | | 5.03 |
| GPT-4V | | 5.26 |
| Gemini-Pro-1.5 | | 5.40 |
| GPT-4o | SoM | 4.59 |
| GPT-4o | AGUVIS-7B | **11.07** |
| | AGUVIS-72B | **10.26** |

In our online experiments, we explore two distinct configurations. The first configuration employs GPT-4o as the planner, collaborating with our AGUVIS-7B, which serves as the grounder. The second setup utilizes our AGUVIS-72B in a dual role, acting as both the planner and the grounder. We compare the performance of these configurations with existing SOTA methods that use GPT-4(o) models as planners. Unlike existing methods that rely on Set-of-Mark (SoM) or textual HTML/AX-Tree information, AGUVIS uses only screenshots as observations and is restricted to pyautogui actions $\mathcal{A}$ in all environments: We set the screenshot viewport to a resolution of $1280 \times 720$ and disabled all actions based on HTML/AXTree selection.

As shown in Table 4 and Table 5, when incorporating the GPT-4o as planner, AGUVIS-7B outperforms existing work in task success rate across various benchmarks. We further adopt our AGUVIS-72B both as the planner and grounder, achieving the best performance on Mind2Web-Live and MobileMiniWob, which demonstrates the advantage potential of employing purely visual agent models for autonomous GUI interactions. By employing AGUVIS-72B as both the planner and the grounder, we achieve the best performance on Mind2Web-Live and MobileMiniWob. This underscores the advantages of utilizing a unified purely visual agent model for autonomous GUI interactions. Furthermore, we observe that our model demonstrates a significant advantage in terms of efficiency costs compared to both closed-source and open-source models (as discussed below), demonstrating that there is considerable potential for applying purely visual agents in real-world online scenarios.

## 4 ANALYSIS

### 4.1 ABLATION

To assess the impact of each stage in the training pipeline of AGUVIS, we conduct ablation experiments. Specifically, we evaluate the performance of the following variants: (a) a model trained without the second stage (planning training), referred to as AGUVIS-G-7B, and (b) a base model, Qwen2-VL (Wang et al., 2024a), without both stages of our specialized training. We report the results of these ablations on two key benchmarks, Multimodal-Mind2Web and AndroidControl, focusing on the step success rate as the evaluation metric (Table 7). The findings show a clear decline

Table 7: Ablation on AGUVIS-7B on MM-Mind2Web and AndroidControl benchmarks. We report the step success rate. We provide a more comprehensive ablation in Appendix E.1

| Settings | ScreenSpot | Multimodal-Mind2Web | | | AndroidControl | |
|---|---|---|---|---|---|---|
| | | Cross-Task | Cross-Website | Cross-Domain | High-Level | Low-Level |
| AGUVIS-7B | 84.4 | 58.5 | 55.4 | 54.8 | 61.5 | 80.5 |
| (a) w/o Stage 2 | 81.8 | 50.9 | 45.2 | 45.3 | 58.0 | 75.6 |
| (b) w/o Stage 1 | 77.4 | 59.7 | 55.3 | 56.8 | 58.8 | 79.8 |
| (c) w/o Stage 1 & 2 | 55.3 | 50.9 | 44.9 | 47.7 | 59.1 | 59.2 |
| (d) w/o Inner Monologue | 79.3 | 55.4 | 53.7 | 54.9 | 60.3 | 69.1 |

in performance when either training stage is omitted. Notably, omitting the second stage (planning and reasoning) has a more significant negative effect on the model's step success rate, indicating that planning training is critical for enhancing the agent's ability to handle complex GUI tasks.

## 4.2 GENERAZATION ON OTHER VLM BACKBONE

Table 8: Performance of AGUVIS based on LLaVA-OneVision backbone. We report the average score on ScreenSpot and the step success rate of each split in Multimoda-Mind2Web. These results demonstrate that our framework and data recipe are model independent and the planning stage can largely improve the performance of both grounding and planning ability.

| Models | ScreenSpot | MM-Mind2Web | | |
|---|---|---|---|---|
| | Average | Task | Website | Domain |
| Previous SOTA | 73.3 | 39.4 | 36.5 | 42.0 |
| AGUVIS$_{ov}$-G-7B | 70.0 | 43.4 | 39.0 | 40.7 |
| AGUVIS$_{ov}$-7B | 81.2 | 55.3 | 50.0 | 50.8 |

Figure 3: Error analysis on Screenspot dataset under the self-plan setting.

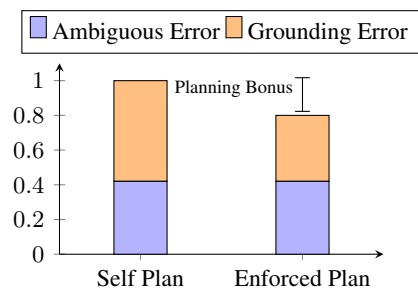

In our experiments, we also implement a version of AGUVIS based on another typical VLM LLaVA-OneVision (Li et al., 2024a), named AGUVIS$_{ov}$-7B, to explore the generalizability of AGUVIS. We report the average score of ScreenSpot and the step success rate of Multimoda-Mind2web. These results demonstrate that our framework and data recipe are model-independent and the planning training stage can largely improve the performance of both grounding and planning ability.

## 4.3 EFFICIENCY

We investigate the efficiency costs of AGUVIS on the online planning benchmark Mind2Web-Live. Following Pan et al. (2024a), we adopt the USD Efficiency Score to evaluate the efficiency of our model in completing tasks. Specifically, this Score is calculated as the total dollar cost of tokens used by the model to complete all tasks in the dataset divided by the total Success Steps. A lower USD Efficiency Score indicates that the model requires fewer USD to complete a successful step. In addition to the USD Efficiency Score, we calculated the number of tokens consumed during the completion of the whole dataset divided by the total number of steps taken by agent models. This reflects the average number of tokens consumed per step.

As shown in Figure 2, AGUVIS significantly reduces the efficiency costs by reducing 93% USD costs and 70% input tokens per step compared to GPT-4o, which indicates considerable potential for applying purely visual agents in practical applications.

## 4.4 ERROR ANALYSIS

We conduct an error analysis of AGUVIS on 50 samples from the ScreenSpot dataset under the self-plan setting to understand the impact of planning on performance. As shown in Figure 3, our findings reveal that 40% of errors are due to ambiguous instructions that could refer to multiple grounding targets, while the remaining 60% are grounding errors. We observe that in these error cases, the

model tends to perform direct grounding action rather than planning explicitly before acting. Notably, when we enforce planning by prompting the agent model to generate low-level instructions before execution, it resolved 20% of the grounding errors. This suggests that while the agent model possesses strong grounding capabilities, there remains significant potential for improvement in effectively leveraging planning and reasoning. These insights highlight opportunities for future work, including improving instruction clarity through the agent model itself, developing adaptive planning mechanisms, and refining training data to include more diverse planning scenarios. Addressing these aspects could further enhance our GUI agent model's robustness on various tasks and environments.

## 5 RELATED WORK

### 5.1 BENCHMARKS AND DATASETS FOR GUI AGENT

Recent advancements in autonomous GUI agents have led to the development of numerous benchmarks and datasets. Web-based benchmarks such as Mind2Web (Deng et al., 2023), WebArena (Zhou et al., 2024; Koh et al., 2024a), WebLINX (Lù et al., 2024), WorkArena (Drouin et al., 2024) and WebCanvas (Pan et al., 2024b) focus on evaluating agents' performance in web environments. For desktop and mobile platforms, datasets like OSWorld (Xie et al., 2024), WindowsAgentArena (Bonatti et al., 2024), AitW (Rawles et al., 2024b), AitZ (Zhang et al., 2024b), AMEX (Chai et al., 2024), GUI-Odyssey (Lu et al., 2024) and AndroidControl (Li et al., 2024b) have been introduced to assess agents' capabilities across different operating systems and device types. Cross-platform datasets such as ScreenSpot (Cheng et al., 2024), OmniACT (Kapoor et al., 2024), GUICourse (Chen et al., 2024a), and CRAB (Xu et al., 2024a) aim to provide comprehensive evaluation frameworks spanning multiple devices and interfaces. Evaluations on specialized applications have also emerged, such as WonderBread (Wornow et al., 2024)'s focus on business process management tasks and Spider-2V (Cao et al., 2024)'s on data science and engineering workflows. In this work, we extensively test benchmarks under both online and offline task settings to thoroughly evaluate and demonstrate the model's planning and grounding capabilities.

### 5.2 MODELS AND APPROACHES FOR GUI AGENT

In parallel with dataset development, significant progress has been made in creating more capable GUI agents. Models like WebGPT (Nakano et al., 2021), Lemur (Xu et al., 2024b), AgentLumos (Yin et al., 2024), CogAgent (Hong et al., 2024), AutoWebGLM (Lai et al., 2024) and xLAM (Zhang et al., 2024a) have demonstrated improved performance in web navigation tasks. Auto-GUI (Zhang & Zhang, 2024), AppAgent (Zhang et al., 2023), and ScreenAgent (Niu et al., 2024) propose novel approaches for direct GUI interaction without relying on application-specific APIs. SearchAgent (Koh et al., 2024b) introduces an inference-time search algorithm to enhance multi-step reasoning and planning in interactive web environments. These advancements collectively contribute to developing more sophisticated and capable GUI agents, pushing the boundaries of what's possible in automated task completion across various digital platforms.

## 6 CONCLUSION

In this paper, we introduced AGUVIS, a unified pure vision-based framework for building autonomous GUI agents that operate across diverse platforms. By only leveraging vision-based observations and a consistent action space, AGUVIS addresses the key challenges of GUI grounding, planning, and reasoning. Our framework unifies and augments existing datasets, enabling more effective cross-platform generalization while reducing inference costs. Extensive experiments demonstrate that AGUVIS outperforms existing methods in both offline and online GUI tasks, showcasing the first fully autonomous pure vision GUI agent capable of completing real-world tasks without reliance on closed-source models. We will open-source all data, models, and training recipes to facilitate future research in this exciting domain.

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

# Table of Contents in Appendix

# A AGUVIS UNIFIED DESIGN

## A.1 DETAILS OF ACTION SPACE IN AGUVIS

In this section, we introduce our unified action space of our pure vision agent framework AGUVIS. As shown in Table 9, we use default standard `pyautogui` actions with pluggable actions as the action space of AGUVIS, which ensures the agent model's universality across environments as well as its flexibility in the specific environment.

Table 9: Default standard `pyautogui` actions $\mathcal{A}$ with pluggable actions.

| Category | Action Space |
|---|---|
| Basic Actions | `pyautogui.moveTo(x, y)`
`pyautogui.click(x, y)`
`pyautogui.write('text')`
`pyautogui.press('enter')`
`pyautogui.hotkey('ctrl', 'c')`
`pyautogui.scroll(200)`
`pyautogui.dragTo(x, y)` |
| Pluggable Actions | `browser.select_option(x, y, value)`
`mobile.swipe(from, to)`
`mobile.home()`
`mobile.back()`
`mobile.open_app(name)`
`terminate(status)`
`answer(text)` |
| ... | ... |

## A.2 PLUGGABLE FUNCTIONS: MOBILE ENVIRONMENTS AS AN EXAMPLE

In the mobile environment, we provide the following pluggable functions for Aguvis, along with their corresponding descriptions as shown in Figure A.2.

---

**Pluggable Functions for AGUVIS**

```
You are a GUI agent. You are given a task and a screenshot of the
screen. You need to perform a series of pyautogui actions to
complete the task.

You have access to the following functions:
- {"name": "mobile.home", "description": "Press the home button"}
- {"name": "mobile.back", "description": "Press the back button"}
- {
    "name": "mobile.long_press",
    "description": "Long press on the screen",
    "parameters": {
        "type": "object",
        "properties": {"x": {"type": "number", "description": "The
        x coordinate of the long press"}, "y": {"type": "number",
        "description": "The y coordinate of the long press"}},
        "required": ["x", "y"]
    }
}
- {
    "name": "mobile.open_app",
    "description": "Open an app on the device",
    "parameters": {
        "type": "object",
```

---

```
            "properties": {"app_name": {"type": "string",
            "description": "The name of the app to open"}},
            "required": ["app_name"]
    }
  }
- {
    "name": "terminate",
    "description": "Terminate the current task and report its
    completion status",
    "parameters": {
        "type": "object",
        "properties": {"status": {"type": "string", "enum":
        ["success"], "description": "The status of the task"}},
        "required": ["status"]
    }
  }
- {
    "name": "answer",
    "description": "Answer a question", "parameters": {
        "type": "object",
        "properties": {"answer": {"type": "string", "description":
        "The answer to the question"}},
        "required": ["answer"]
    }
  }
```

## B  DATA CURATION OF THE AGUVIS COLLECTION

### B.1  DETAILED SOURCE DATASET STATISTICS

We present the detailed statistical information of all training datasets utilized in both the grounding and planning & reasoning stages. The statistics are shown in Table 10 and Table 11, respectively.

Table 10: The grounding split of THE AGUVIS COLLECTION. Each example in this split consists of a single-step trajectory.

| Data source | Platform | Instruction | #Trajectory |
|---|---|---|---|
| SeeClick (Cheng et al., 2024) | Website | Augmented | 271K |
| GUIEnv (Chen et al., 2024a) | Website | Augmented | 328K |
| GUIAct (Chen et al., 2024a) | Website | Original | 67K |
| WebUI (Wu et al., 2023) | Website | Augmented | 57K |
| Widget Captioning (Li et al., 2020b) | Mobile | Original | 101K |
| RicoSCA (Li et al., 2020a) | Mobile | Original | 173K |
| UI RefExp (Bai et al., 2021) | Mobile | Original | 16K |
| RICO Icon (Deka et al., 2017) | Mobile | Augmented | 16K |
| OmniACT (Kapoor et al., 2024) | Desktop & Website | Original | 7K |
| Total | | | 1.036M |

### B.2  PROMPT FOR AUGMENTING PLANNING & REASONING TRAJECTORIES

> **Prompt for GPT-4o generating planning & reasoning data**
>
> ```
> Goal: {goal}
> Previous Actions: {previous_actions}
>
> Given the current screenshot and the next ground truth action
> labeled as `{current_action_instruction}`, the action commands is:
> ```

```json
{action_commands}
```

This element is highlighted in red bounding box in the image.

Describe the situation in detail, focusing on the goal and current
observation. Ensure your reasoning aligns with the goal and the
labeled action, but avoid using the labeled action or the
highlighted bounding box as reasoning support, as they represent
hindsight rather than predictive insight. Conclude with a clear,
actionable instruction in one sentence. Aim to reason through the
task as if solving it, rather than simply reflecting on the labeled
outcome. Use the first-person perspective to represent the
annotator's thought process.

We use GPT-4o as the foundational model to augment our integrated agent trajectory. In this stage, goal represents the target of the trajectory, previous_actions is a stack of all past low-level instructions, current_action_instruction refers to the low-level instruction corresponding to the current action in the dataset, and action_commands is the representation of the current action in the form of PyAutoGUI code within the dataset.

### B.3 HUMAN STUDY ON AUGMENTED DATA

#### B.3.1 QUALITATIVE HUMAN STUDY

Based on our findings that our Augmented Planning and Reasoning Data improves the performance of Aguvis, we conducted a qualitative study on augmented data. From the VLM-augmented data, we selected 90 samples for a human study and evaluated them according to specific criteria.

We determined that for augmented data to be considered successful, it must:

- Match the action type and action target elements of the ground truth,
- Correctly describe the step's intention,
- Establish a clear connection between the step's intention and the overall goal,
- Assist the agent in successfully completing the task.

Among the sampled data, we found that 86.7% demonstrated intermediate reasoning that aligned with the ground truth actions and the overall goal's action intention. The remaining 7.8% cases were influenced by dataset noise (irrelevant or unnecessary actions within the task), and 5.5% cases were due to misinterpretations of the action intention under clean data.

#### B.3.2 FAILURE CASES UNDER NOISY TRAINING DATA

We analyzed error cases in the generated data and identified several issues. Specifically, we found that unnecessary actions in the training data can lead to the VLM failing to establish a connection

Table 11: The planning & reasoning split of THE AGUVIS COLLECTION.

| Data source | Platform | Inner Monologue | Avg. Steps | #Trajectory |
|---|---|---|---|---|
| MM-Mind2Web (Zheng et al., 2024a) | Website | Generated | 7.7 | 1,009 |
| GUIAct (Chen et al., 2024a) | Website | Generated | 6.7 | 2,482 |
| MiniWoB++ (Zheng et al., 2024b) | Website | Generated | 3.6 | 2,762 |
| AitZ (Zhang et al., 2024b) | Mobile | Original | 6.0 | 1,987 |
| AndroidControl (Li et al., 2024d) | Mobile | Original | 5.5 | 13,594 |
| GUI Odyssey (Lu et al., 2024) | Mobile | Generated | 15.3 | 7,735 |
| AMEX (Chai et al., 2024) | Mobile | Generated | 11.9 | 2,991 |
| AitW (Rawles et al., 2024b) | Mobile | Generated | 8.1 | 2,346 |
| Total | | | | 35K |

between these extra actions and the overall goal, ultimately resulting in incorrect reasoning and planning.

While these redundant actions do not compromise the trajectory's overall completeness or correctness, they do introduce challenges for the VLM in generating accurate planning.

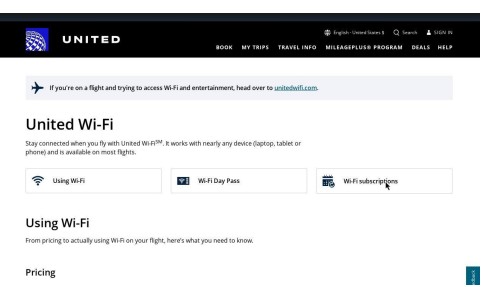
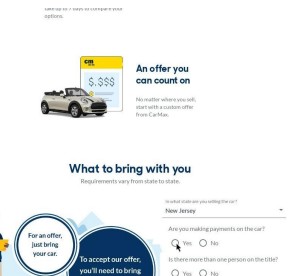

**Goal:** Show me the page with information about Wi-Fi subscriptions.

**Goal:** Find the documents required to sell a car in New Jersey for which payments are being paid and it has single person on the title who can come to the store

**Previous actions:**
Step 1: Click on the 'TRAVEL INFO' tab to explore available options for Wi-Fi subscription information.
Step 2: Click on the 'Pre-paid Wi-Fi' link to access Wi-Fi subscription information.

**Previous actions:**
Step 1: Click on the 'Sell/Trade' link in the navigation bar to proceed.
Step 2: Select 'New Jersey' from the dropdown menu labeled 'In what state are you selling the car?'.

**Action:** pyautogui.click(x=0.8018, y=0.6183)

**Action:** pyautogui.click(x=0.5938, y=0.8633)

Generated Planning

Generated Planning

**Observation:** I am on a page related to United Wi-Fi options, which includes a link for Wi-Fi subscriptions. This is in line with my goal to gather information about Wi-Fi subscriptions for flights.

**Observation:** The page asks if payments are still being made on the car and if there is more than one person on the title, with the option to select either 'Yes' or 'No' for both questions. Currently, the focus is on confirming that payments are being made, as illustrated by the page.

**Thought:** Identifying the 'Wi-Fi subscriptions' link on the United Wi-Fi page aligns perfectly with the goal of finding subscription information. I am considering available options and see a dedicated link for subscriptions.

**Thought:** To determine the document requirements for selling a car in New Jersey, I need to select options that accurately reflect the current status of the car's payments and ownership. Since payments are being made, I will choose 'Yes' for that question to proceed and assess any further requirements or conditions this might bring into play.

**Low-level Instruction:** Click on the 'Wi-Fi subscriptions' link to access information about Wi-Fi subscriptions.

**Low-level Instruction:** Click on the 'Yes' radio button next to 'Are you making payments on the car?'

Figure 4: Examples of augmented planning and reasoning data generated by GPT-4o. The position of the mouse in the image represents the ground truth click position in the training data.

## C  AGUVIS TRAINING

### C.1  TRAINING EXAMPLE SCHEMA

---

**Training Data Schema of Stage 1 Grounding**

**Prompt**

```
<|im_start|>system
You are a GUI agent. You are given a task and a screenshot of the
screen. You need to perform a series of pyautogui actions to
complete the task.<|im_end|>
<|im_start|>user
<|vision_start|><|image_pad|><|vision_end|>
Please generate the next move according to the ui screenshot,
instruction and previous actions.
Instruction: {overall_goal}
Previous actions: {previous_actions}
<|im_end|>
```

---

**Generation**

```
<|im_start|>assistant<|recipient|>os
Action: {pyautogui function}
<|diff_marker|>
```

---

**Training Data Schema of Stage 2 Planning**

**Prompt**

```
<|im_start|>system
You are a GUI agent. You are given a task and a screenshot of the
screen. You need to perform a series of pyautogui actions to
complete the task.<|im_end|>
<|im_start|>user
<|vision_start|><|image_pad|><|vision_end|>
Please generate the next move according to the ui screenshot,
instruction and previous actions.
Instruction: {overall_goal}
Previous actions: {previous_actions}
<|im_end|>
```

---

**Generation**

```
<|im_start|>assistant<|recipient|>all
Observation: {Observation}
Thought: {Planning}
Low-level Instruction: {Low-level Instruction}
<|im_end|>
<|im_start|>assistant<|recipient|>os
Action: {pyautogui function}
<|diff_marker|>
```

---

AGUVIS introduces a novel explicit planning and reasoning training framework that differs from existing approaches. We illustrate these differences with visual examples in Figure 5. While existing training datasets utilize trajectory data to fine-tune agents, these approaches often involve agents directly outputting action commands (e.g., via pyautogui), bypassing the generation of observations, thoughts, and low-level instructions in natural language that correspond to actions. To elicit the reasoning and planning capabilities of vision-language models and provide the model with richer context for action generation, we scale up training datasets that explicitly require the model to output

reasoning and planning steps. Moreover, this approach enhances the interpretability of computer-use agents' behavior, laying a solid foundation for future research.

## C.2 TRAINING DETAILS

For AGUVIS based on the Qwen2-VL backbone, we set the maximum pixels for each image to $1280 \times 720$ to achieve a better trade-off between performance and efficiency[1]. Following the SFT strategy in Wang et al. (2024a), we freeze the ViT parameters during training. For AGUVIS based on the LLaVA-OneVision backbone, we adopt the *anyres* strategy, which splits high-resolution images into multiple patches following (Li et al., 2024a). The maximum sequence length of tokens is set to 8192 for all models. We use Adam optimizer (Loshchilov & Hutter, 2019) for both grounding and planning & reasoning training stages and employ a cosine learning rate scheduler with a warm-up ratio of 3% steps. In the grounding stage, we introduce a grounding packing strategy to enhance training efficiency. We conduct an ablation study using the grounding data of website platform to investigate the strategy effectiveness. We observe that it reduces overall GPU hours from 6 hours to 1 hour. Moreover, this strategy even marginally improve the performance of ScreenSpot website split from 73.3 to 76.8.

We train AGUVIS with a batch size of 128 for 1 epoch in each stage. The peak learning rate is set to 1e-5 for AGUVIS-7B and 5e-6 for AGUVIS-72B. Our codebase is based on Pytorch (Paszke et al., 2019) and Huggingface Transformers (Wolf et al., 2019). During training, we utilize the strategies of DeepSpeed optimization (Rajbhandari et al., 2020), BF16 format and gradient checkpointing to save GPU memory. We train AGUVIS on a cluster of H100-80G GPUs: AGUVIS-7B uses 8 nodes and completes the grounding training within 5 hours and planning & reasoning training within 1 hour. AGUVIS-72B uses 16 nodes and completes the grounding training within 30 hours and planning & reasoning training within 6 hours.

## D EVALUATION BENCHMARKS

In this section, we introduce more details of evaluation benchmarks used in our work.

### D.1 GUI GROUNDING EVALUATION

**ScreenSpot.** ScreenSpot (Cheng et al., 2024)is a typical benchmark designed specifically for GUI visual grounding, consisting of 1.2K single-step instructions and coordinates of the target elements. This dataset encompasses a variety of grounding instructions tailored for mobile, desktop, and website platforms, and categorizes element types into text and icons/widgets. The benchmark is assessed under two distinct settings: (1) *Original Instructions*: models perform grounding actions directly following the original instructions; and (2) *Self-plan*: models are required to generate plans in natural language based on the original instructions before executing grounding actions.

### D.2 OFFLINE GUI AGENT EVALUATION

**Multimodal-Mind2Web.** We utilize Multimodal-Mind2Web (Zheng et al., 2024a) for evaluating the offline planning capabilities of GUI agents on websites, which builds on the original Mind2Web (Deng et al., 2023). We report element accuracy (Ele.Acc), Operation F1 (Op.F1), and step success rate (Step SR).

In Table 2 for Multimodal Mind2Web (Zheng et al., 2024a), we only report element accuracy for SeeClick (Cheng et al., 2024) and CogAgent (Hong et al., 2024). This is because the original SeeClick and CogAgent models were evaluated on Mind2Web (Deng et al., 2023), not Multimodal Mind2Web, making the examples misaligned and incomparable. Therefore, we referenced the results from UGround (Gou et al., 2024), where they report the element accuracy of the SeeClick and CogAgent models on Multimodal Mind2Web, striving to comprehensively present all previously representative methods.

---

[1]During preliminary experiments, we observe that increasing the maximum pixels to $1920 \times 1080$ does not yield significant improvements on ScreenSpot performance.

**Image Input**

**Existing GUI Agent Data**

**Aguvis Collection Data**

**Prompt**

Please generate the next move according to the UI screenshot, instruction and previous actions.

**Instruction:** Plan a trip from Boston Logan Airport to North Station.

**Previous actions:**
Step 1: pyautogui.click(x=0.4754, y=0.2062)
Step 2: pyautogui.click(x=0.3295, y=0.4)
pyautogui.write(text='Boston Logan Airport')
Step 3: pyautogui.click(x=0.3262, y=0.4764)

**Generation**

**Prompt**

Please generate the next move according to the UI screenshot, instruction and previous actions.

**Instruction:** Plan a trip from Boston Logan Airport to North Station.

**Previous actions:**
Step 1: Click on the 'Trip Planner' tab to begin planning the trip.
Step 2: Click on the 'From' input field and type 'Boston Logan Airport'.
Step 3: Click on 'Boston Logan Int'l Airport, 1 Harborside Dr, East Boston, MA 02128, United States' to set it as my starting location.

**Generation**

**Observation:** The trip planner interface is open with the 'From' field set to Boston Logan Airport. The 'To' field is empty, awaiting input for the destination.

**Thought:** I have set my starting point as Boston Logan Airport. To proceed, I need to set the destination to North Station, allowing the trip planner to suggest routes.

**Low-level Instruction:** Click on the 'To' input field and type 'North Station' as the destination.

**Action:**
pyautogui.click(x=0.6756, y=0.4)
pyautogui.write(text='North Station')

**Action:**
pyautogui.click(x=0.6756, y=0.4)
pyautogui.write(text='North Station')

Figure 5: Compared to the schema of exisiting gui agent data (left), the schema of AGUVIS planning & reasoning data (right) includes explicit reasoning process with informative natural languaeg previous action context.

**AndroidControl.** Following the setting in Li et al. (2024d), we randomly sample 500 step-actions from AndroidControl full test set to create a subset, and we report the step accuracy on out-of-domain (OOD) data within both high-level and low-level tasks. The high-level task setting necessitates that the model plans and executes actions, whereas the low-level task setting requires the model to simply adhere to human-labeled instructions for executing the next-step action.

### D.3 ONLINE GUI AGENT EVALUATION

**Mind2Web-Live.** We adopt Mind2Web-Live (Pan et al., 2024b) to evaluate GUI agents' online planning, a derived dynamic data set from Mind2Web, comprising 104 real-time interactive web tasks. It evaluates whether each required step within a task has been successfully completed and uses the task success rate (Task SR) as the reported metric. The original Mind2Web-Live is built with WebCavas (Pan et al., 2024a), which is a text-based agent framework. To better accommodate the unified observation and action space of pure vision models, we utilize BrowserGym (Drouin et al., 2024) as the evaluation environment for online web tasks which provide support for pure vision-based agent models. BrowserGym is a browser testing environment built on the Playwright (Microsoft, 2024) engine. We incorporate all Mind2Web-Live tasks and evaluation into BrowserGym, involving registering all Mind2Web-Live tasks, setting up the entry points for these tasks, and porting the Mind2Web-Live evaluation functions to BrowserGym.

As Mind2Web-Live is a text-based benchmark, we have to adapt its evaluation function to suit our pure vision-based model. To achieve this, we introduce the two modifications following:

- For the Mind2Web-Live benchmark's click verification, we adapt our coordinate-based approach by comparing the ground truth CSS selector's bounding box (when available) with our click coordinates, as we cannot directly identify HTML elements.

- Similarly, for input validation, we retrieve and compare the value of the ground truth input element (if present) with the expected value, circumventing the need for precise HTML element identification based on CSS selectors.

The Mind2Web-Live environment relies on real-world websites, many of which implement detection systems for automated browser testing and reCAPTCHA challenges. These factors created difficulties during evluation on the Mind2Web-Live dataset, resulting in a lower task success rate (Task SR). Specifically, we observed the following websites to have significant issues with automation detection:

- **kohls**. Model using the search functionality on the Kohls website through Playwright directly results in a 502 Bad Gateway error.

- **target**. We are unable to open target's job website using Playwright due to network connection error.

- **united**. We are unable to open united website using Playwright due to network connection error.

In addition to the websites that were consistently prone to failure, several other sites intermittently blocked our Playwright access during testing. In total, we encountered 18 network errors and 6 reCAPTCHA tasks that the model was unable to complete, preventing our model from scoring on these 24 tasks.

**AndroidWorld.** AndroidWorld (Rawles et al., 2024b) is a benchmark operating on an Android virtual environment, capable of dynamically instantiating with randomly generated parameters to generate unique tasks for automatic evaluation. It spans 20 real-world applications, encompassing 116 diverse tasks. To assess the pure vision agent models, we follow the instructions in Rawles et al. (2024b), installing a Pixel 6 phone simulator on our computers to serve as the experimental environment. The benchmark incorporates a fully automated task-level evaluation system that automatically assesses whether a state has successfully completed a designated task. The AndroidWorld environment supports optional inputs such as Set-of-Mark (SoM) and textual AXTree information, which most multimodal models currently rely on to complete tasks. However, we solely use raw screenshots as the observation input and restrict the model to coordinate-level actions and basic mobile functions.

**MobileMiniWob.** MobileMiniWob (Rawles et al., 2024b) is the instantiation of 92 tasks from MiniWob++ (Zheng et al., 2024b) in the AndroidWorld environment. Thus, we adopt the same observation and action space used in AndroidWorld and use a real-time evaluation function to determine task success.

### D.3.1 PROMPTS FOR USING GPT-4O AS PLANNING MODEL

In all online experiments, we employed two settings: GPT-4o as the planner, AGUVIS-7B as the grounder, and AGUVIS-72B as both the planner and grounder. For experiments where AGUVIS-72B served as both the planner and grounder, the prompt was straightforward: we only needed to provide AGUVIS-72B with a single prompt at each step, and it could independently handle reasoning, planning, and grounding. We use prompt for forcing plan to improve AGUVIS-72B's performance on the online experiments, as illustrated in Appendix E.2.2

In the GPT-4o + AGUVIS-7B setting, the situation was more complex. Two key challenges needed to be addressed: making GPT-4o's planning usable by AGUVIS-7B and determining which actions required AGUVIS-7B for grounding. To address these challenges, we modified GPT-4o's prompts based on Mind2Web-Live (BrowserGym) and AndroidWorld to enable it to delegate grounding actions to AGUVIS-7B when necessary and to share its planning outputs with AGUVIS-7B. Specifically, we append `<|im_start|>assistant<|recipient|>all\nThought:{GPT-4o Thought}\nAction:{GPT-4o Low-level Instruction}` to the end of the prompt and therefore let AGUVIS-7B generate grounding actions based on GPT-4o's response.

Table 12: Prompt used for the planning model in **Mind2Web-Live**, modified from the prompt in (Drouin et al., 2024)

---

**Instructions**
Review the current state of the page and all other information to find the best possible next action to accomplish your goal. Your answer will be interpreted and executed by a program, make sure to follow the formatting instructions.

---

**Goal**: {Goal}

---

**Observation of current step**
Current URL: {URL}
History of interaction with the task: {History}

---

**Action Space**
8 different types of actions are available.

noop(wait_ms: float = 1000)
Description: Do nothing, and optionally wait for the given time (in milliseconds).

send_msg_to_user(text: str)
Description: Sends a message to the user.

scroll(delta_x: float, delta_y: float, relative: bool = False)
Description: Scroll horizontally and vertically. Amounts in pixels, positive for right or down scrolling, negative for left or up scrolling. Dispatches a wheel event.

fill(element: str, value: str)
Description: Fill out a form field. It focuses the element and triggers an input event with the entered text. It works for <input>, <textarea>, and [contenteditable] elements. The 'element' parameter represents the semantic information of the element you want to fill.

click(element: str, button: Literal['left', 'middle', 'right'] = 'left')
Description: Click an element. The 'element' parameter represents the semantic information of the element you want to click.

dblclick(element: str, button: Literal['left', 'middle', 'right'] = 'left')

---

*Continued on the next page*

Table 12 – Continued from the previous page

**Instructions**

Review the current state of the page and all other information to find the best possible next action to accomplish your goal. Your answer will be interpreted and executed by a program, make sure to follow the formatting instructions.

Description: Double click an element. The 'element' parameter represents the semantic information of the element you want to double click.

hover(element: str)
Description: Hover over an element. The 'element' parameter represents the semantic information of the element you want to hover over.

keyboard_press(key: str)
Description: Press a combination of keys. Accepts the logical key names that are emitted in the keyboardEvent.key property of the keyboard events: Backquote, Minus, Equal, Backslash, Backspace, Tab, Delete, Escape, ArrowDown, End, Enter, Home, Insert, Page-Down, PageUp, ArrowRight, ArrowUp, F1 - F12, Digit0 - Digit9, KeyA - KeyZ, etc. You can alternatively specify a single character you'd like to produce such as "a" or "#". Following modification shortcuts are also supported: Shift, Control, Alt, Meta.

Only a single action can be provided at once. Example:
fill('comment text area', 'This is an example')
Note: you are on mac so you should use Meta instead of Control for Control+C etc.

Table 13: Prompts used for the planning model in **AndroidWorld**, modified from the prompt in (Rawles et al., 2024a)

**Instruction**

You are an agent who can operate an Android phone on behalf of a user. Based on user's goal/request, you may
- Answer back if the request/goal is a question (or a chat message), like user asks "What is my schedule for today?".
- Complete some tasks described in the requests/goals by performing actions (step by step) on the phone.

When given a user request, you will try to complete it step by step. At each step, you will be given the current screenshot and a history of what you have done (in text). Based on these pieces of information and the goal, you must choose to perform one of the action in the following list (action description followed by the JSON format) by outputting the action in the correct JSON format.
- If you think the task has been completed, finish the task by using the status action with complete as goal_status: '{"action_type": "status", "goal_status": "complete"}'
- If you think the task is not feasible (including cases like you don't have enough information or can not perform some necessary actions), finish by using the 'status' action with infeasible as goal_status: '{"action_type": "status", "goal_status": "infeasible"}'
- Answer user's question: '{"action_type": "answer", "text": "answer_text"}'
- Click/tap on an element on the screen. Please describe the element you want to click using natural language. '{"action_type": "click", "target": target_element_description}'.
- Long press on an element on the screen, similar with the click action above, use the semantic description to indicate the element you want to long press: '{"action_type": "long_press", "target": target_element_description}'.
- Type text into a text field (this action contains clicking the text field, typing in the text and pressing the enter, so no need to click on the target field to start), use the semantic description to indicate the target text field: '{"action_type": "input_text", "text": text_input, "target": target_element_description}'

| Table 13 – Continued from the previous page |
|---|

- Press the Enter key: '{"action_type": "keyboard_enter"}'
- Navigate to the home screen: '{"action_type": "navigate_home"}'
- Navigate back: '{"action_type": "navigate_back"}'
- Scroll the screen or a scrollable UI element in one of the four directions, use the same semantic description as above if you want to scroll a specific UI element, leave it empty when scroll the whole screen: '{"action_type": "scroll", "direction": up, down, left, right, "element": optional_target_element_description}'
- Open an app (nothing will happen if the app is not installed): '{"action_type": "open_app", "app_name": name}'
- Wait for the screen to update: '{"action_type": "wait"}'

**Guidelines**
Here are some useful guidelines you need to follow:
General:
- Usually there will be multiple ways to complete a task, pick the easiest one. Also when something does not work as expected (due to various reasons), sometimes a simple retry can solve the problem, but if it doesn't (you can see that from the history), SWITCH to other solutions.
- Sometimes you may need to navigate the phone to gather information needed to complete the task, for example if user asks "what is my schedule tomorrow", then you may want to open the calendar app (using the 'open_app' action), look up information there, answer user's question (using the 'answer' action) and finish (using the 'status' action with complete as goal_status).
- For requests that are questions (or chat messages), remember to use the 'answer' action to reply to user explicitly before finish! Merely displaying the answer on the screen is NOT sufficient (unless the goal is something like "show me ...").
- If the desired state is already achieved (e.g., enabling Wi-Fi when it's already on), you can just complete the task.
Action Related:
- Use the 'open_app' action whenever you want to open an app (nothing will happen if the app is not installed), do not use the app drawer to open an app unless all other ways have failed.
- Use the 'input_text' action whenever you want to type something (including password) instead of clicking characters on the keyboard one by one. Sometimes there is some default text in the text field you want to type in, remember to delete them before typing.
- For 'click', 'long_press' and 'input_text', the target_element_description parameter you choose must based on a VISIBLE element in the screenshot.
- Consider exploring the screen by using the 'scroll' action with different directions to reveal additional content.
- The direction parameter for the 'scroll' action can be confusing sometimes as it's opposite to swipe, for example, to view content at the bottom, the 'scroll' direction should be set to "down". It has been observed that you have difficulties in choosing the correct direction, so if one does not work, try the opposite as well.
Text Related Operations:
- Normally to select certain text on the screen: (i) Enter text selection mode by long pressing the area where the text is, then some of the words near the long press point will be selected (highlighted with two pointers indicating the range) and usually a text selection bar will also appear with options like 'copy', 'paste', 'select all', etc. (ii) Select the exact text you need. Usually the text selected from the previous step is NOT the one you want, you need to adjust the range by dragging the two pointers. If you want to select all text in the text field, simply click the 'select all' button in the bar.
- At this point, you don't have the ability to drag something around the screen, so in general you can not select arbitrary text.

*Continued on the next page*

Table 13 – Continued from the previous page

- To delete some text: the most traditional way is to place the cursor at the right place and use the backspace button in the keyboard to delete the characters one by one (can long press the backspace to accelerate if there are many to delete). Another approach is to first select the text you want to delete, then click the backspace button in the keyboard.
- To copy some text: first select the exact text you want to copy, which usually also brings up the text selection bar, then click the 'copy' button in bar.
- To paste text into a text box, first long press the text box, then usually the text selection bar will appear with a 'paste' button in it.
- When typing into a text field, sometimes an auto-complete dropdown list will appear. This usually indicating this is a enum field and you should try to select the best match by clicking the corresponding one in the list.

# E ANALYSIS

## E.1 TRAINING ABLATION

### E.1.1 TRAINING STRATEGY ABLATION

To further demonstrate the contribution of Stage 1, Stage 2, and their combination to model training, we conducted an ablation study. Specifically, we designed five experimental settings on AGUVIS$_{\text{QWEN2-VL}}$ and AGUVIS$_{\text{LLAVA-OV}}$. We further explain the meaning of each setting:

- **Stage 1 → Stage 2** corresponds to the staged configuration AGUVIS used in our paper, where Stage 1 is followed by Stage 2 sequentially.

- **Stage 1 + Stage 2** represents a joint training setup, where two stages are combined into a training process.

- **w/o Stage x** indicates the absence of the respective stage in the setting.

Note that for each setting, the model is fine-tuned on the corresponding task-specific dataset.

From the first two rows in Table 14, it can be observed that the differences between models trained with Staged Training and Joint Training setups are relatively minor. However, a clear trend emerges: models trained using the Joint Training setup perform better on GUI grounding tasks but exhibit inferior performance on datasets requires planning ability such as MM-Mind2Web and AndroidControl High-level. This trend implies grounding data in Stage 1 is more abundant, dominating the optimization process and biasing the model toward grounding tasks. In contrast, the data in Stage 2, which combines planning and grounding, is of higher quality and better aligned with the agent's deployment scenarios. This rationale underpins our decision to position Stage 2 later in the training sequence.

Moreover, it is observed that compared to AGUVIS$_{\text{QWEN2-VL}}$ trained through both Stage 1 and Stage 2, the model trained with only Stage 2 data maintains similar performance on MM-Mind2Web and AndroidControl but exhibits a notable decline in GUI grounding performance on ScreenSpot. This suggests that the stability on Mind2Web and AndroidControl can be attributed to Qwen2VL's pre-training on natural image grounding. However, the diverse image and domain requirements of the ScreenSpot GUI grounding test set highlight the necessity of extensive and varied grounding training from Stage 1. This training is essential for improving the grounding performance required for a cross-platform GUI agent model.

To verify this analysis, we conduct the same ablation study on the LLaVA model, as shown in Table 15. From the results, we can see that the original LLaVA did not undergo extensive natural image grounding training during the training process, making it insufficient for LLaVA to excel when only Stage 1 or Stage 2 is conducted. When both Stage 1 and Stage 2 are performed, LLaVA can be significantly improved, even surpassing previous SOTA results. This validates the above analysis and further demonstrates that our method is model-agnostic and universally applicable to popular VLMs like Qwen2-VL and LLaVA.

Table 14: Ablation study of AGUVIS$_{\text{QWEN2-VL}}$ on training strategy.

| Settings | ScreenSpot | Multimodal-Mind2Web | | | AndroidControl | |
|---|---|---|---|---|---|---|
| | | Cross-Task | Cross-Website | Cross-Domain | High-Level | Low-Level |
| Stage 1 → 2 | 84.4 | 58.5 | 55.4 | 54.8 | 61.5 | 80.5 |
| Stage 1 + 2 | 85.0 | 56.1 | 53.1 | 55.6 | 59.2 | 80.9 |
| w/o Stage 2 | 81.8 | 50.9 | 45.2 | 45.3 | 58.0 | 75.6 |
| w/o Stage 1 | 77.4 | 59.7 | 55.3 | 55.8 | 58.8 | 79.8 |
| w/o Stage 1 & 2 | 55.3 | 50.9 | 44.9 | 47.7 | 59.1 | 59.2 |

Table 15: Ablation study of AGUVIS$_{\text{LLAVA-OV}}$ on training strategy.

| Settings | ScreenSpot | Multimodal-Mind2Web | | | AndroidControl | |
|---|---|---|---|---|---|---|
| | | Cross-Task | Cross-Website | Cross-Domain | High-Level | Low-Level |
| Stage 1 → 2 | 81.2 | 55.3 | 50.0 | 50.8 | 60.7 | 82.4 |
| w/o Stage 2 | 70.0 | 43.4 | 39.0 | 40.7 | 54.9 | 65.6 |
| w/o Stage 1 | 71.3 | 42.5 | 40.3 | 42.8 | 61.4 | 80.5 |
| w/o Stage 1 & 2 | 3.8 | 33.8 | 30.5 | 32.4 | 50.4 | 50.0 |

### E.1.2 DATA STRATEGY ABLATION

To investigate the impact of different device domain datasets within a unified action space, we designed three settings on the MM-Mind2Web dataset: (1) training with the complete dataset comprising both Web and Mobile data, (2) training using only the Web data, and (3) fine-tuning exclusively on the MM-Mind2Web dataset. All three experiments include fine-tuning on the MM-Mind2Web dataset.

Table 16: Ablation Study of The Impact of Mobile Data on MM-Mind2Web

| Model | Training Data | MM-Mind2Web | | |
|---|---|---|---|---|
| | | Cross-Task | Cross-Website | Cross-Domain |
| AGUVIS$_{\text{QWEN2-VL}}$ | Web + Mobile (Stage 2 Equivalent) | 58.5 | 55.4 | 54.8 |
| | Web Only | 53.1 | 50.3 | 52.2 |
| | Mind2Web Only | 50.9 | 44.9 | 47.7 |
| AGUVIS$_{\text{LLAVA-OV}}$ | Web + Mobile (Stage 2 Equivalent) | 55.3 | 50.0 | 50.8 |
| | Web Only | 44.9 | 43.5 | 42.1 |
| | Mind2Web Only | 43.4 | 39.0 | 40.7 |

Table 17: Ablation Study of the Impact of Inner Monologue

| AGUVIS | ScreenSpot | Multimodal-Mind2Web | | | AndroidControl | |
|---|---|---|---|---|---|---|
| | | Cross-Task | Cross-Website | Cross-Domain | High-Level | Low-Level |
| AGUVIS | 84.4 | 58.5 | 55.4 | 54.8 | 61.5 | 80.5 |
| AGUVIS w/o IM | 79.3 | 55.4 | 53.7 | 54.9 | 60.3 | 69.1 |

The experimental results, presented in the Table 16, demonstrate that training AGUVIS with both Web and Mobile data consistently outperforms the setting trained exclusively on MM-Mind2Web. This performance gain underscores the contribution of Mobile data to enhancing cross-device domain generalization in the Web domain, validating the effectiveness of our cross-platform data.

In addition, we conducted ablation study on the role of incorporating inner monologue (IM) in training. The result shown in Table 17 demonstrated clear performance gain from inner monologue. This gain can be attributed to two key factors: the use of inner monologue enables the model to elicit reasoning about the current step while also serving as context to facilitate more effective planning

for subsequent steps. Additionally, incorporating low-level instructions from the training data improves the accuracy of the model's action execution, as demonstrated in both the Screenspot and AndroidControl low-level tasks.

### E.2 PLANNING ANALYSIS

#### E.2.1 PROMPTS FOR SELF-PLANNING AND ENFORCED PLANNING MODE.

In Appendix C.1, we present the training data schema for Stage 1 and Stage 2. We use the special token `<|recipient|>` along with `os` or `all` to control whether the message content is an inner monologue or a pyautogui action command. Thanks to this design, we can use `<|recipient|>` during the inference phase to control the content generated by the model.

In the Enforced Plan Setting, we employ the `<|recipient|>all\nThought` prompt to compel the model to generate a planning phase following this. While in the self-plan setting, we do not add any word after `<|recipient|>`, so the model can choose to generate `os` to directly produce a pyautogui command, or generate `all` to first create natural language reasoning and then generate a pyautogui command.

---

**Prompt Template For Self-plan**

```
<|im_start|>system
You are a GUI agent. You are given a task and a screenshot of the
screen. You need to perform a series of pyautogui actions to
complete the task.<|im_end|>
<|im_start|>user
<|vision_start|><|image_pad|><|vision_end|>Please generate the next
move according to the ui screenshot, instruction and previous
actions.

Instruction: {goal}

Previous actions: {previous_actions}
<|im_end|>
<|im_start|>assistant<|recipient|>
```

---

**Prompt Template For Enforced Plan**

```
<|im_start|>system
You are a GUI agent. You are given a task and a screenshot of the
screen. You need to perform a series of pyautogui actions to
complete the task.<|im_end|>
<|im_start|>user
<|vision_start|><|image_pad|><|vision_end|>Please generate the next
move according to the ui screenshot, instruction and previous
actions.

Instruction: {overall_goal}

Previous actions: {previous_actions}
<|im_end|>
<|im_start|>assistant<|recipient|>all
Thought:
```

## E.2.2 PLANNING BOUNUS EXAMPLES

**Task Instruction: add a new file**

pyautogui.click(x=0.3997, y=0.0148) ❌

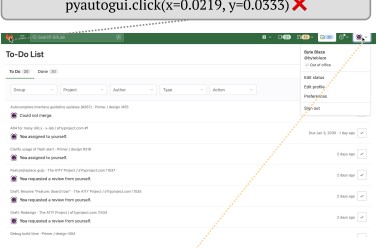

**Thought:** To add a new file, I need to select the 'New' option from the context menu. This will likely open a dialog or prompt to create a new file, aligning with the goal of adding a new file to the directory.
**Action:** Click on the 'New' option in the context menu.
pyautogui.click(x=0.6836, y=0.2535) ✅

**Task Instruction: send current page**

pyautogui.click(x=0.9792, y=0.8046) ❌

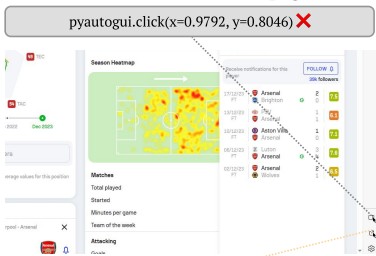

**Thought:** To share the current page, I need to find and click the share icon, which is typically represented by a network or link symbol. This icon is usually located in the browser's toolbar or menu.
**Action:** Click the share icon in the browser to share the current page.
pyautogui.click(x=0.9792, y=0.876) ✅

**Task Instruction: go to personal homepage**

pyautogui.click(x=0.0219, y=0.0333) ❌

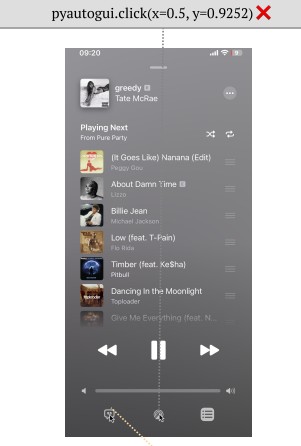

**Thought:** To reach the personal homepage, I need to interact with the user profile element, which is typically represented by an avatar or username. This element is usually clickable and leads to the user's profile page.
**Action:** Click on the user profile icon or username at the top right corner to go to the personal homepage.
pyautogui.click(x=0.9615, y=0.0326) ✅

**Task Instruction: rate this product with four stars**

pyautogui.click(x=0.2271, y=0.2926) ❌

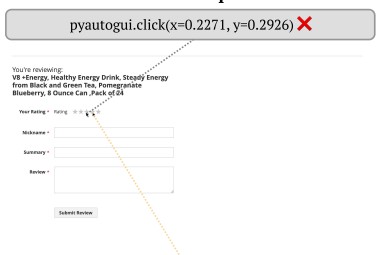

**Thought:** To achieve the goal of rating the product with four stars, I need to click on the fourth star in the rating section. This will ensure the product is rated with four stars, aligning with the task requirement.
**Action:** Click on the fourth star in the rating section to rate the product with four stars.
pyautogui.click(x=0.2445, y=0.293) ✅

**Task Instruction: switch to song lyric**

pyautogui.click(x=0.5, y=0.9252) ❌

**Thought:** To switch to song lyrics, I need to find and click on the lyrics button, which is typically represented by a text or chat icon. This icon is usually located at the bottom of the screen among other playback controls.
**Action:** Click on the lyrics icon at the bottom of the screen to switch to song lyrics.
pyautogui.click(x=0.2352, y=0.9267) ✅

**Task Instruction: change dark mode options**

pyautogui.click(x=0.7306, y=0.3575) ❌

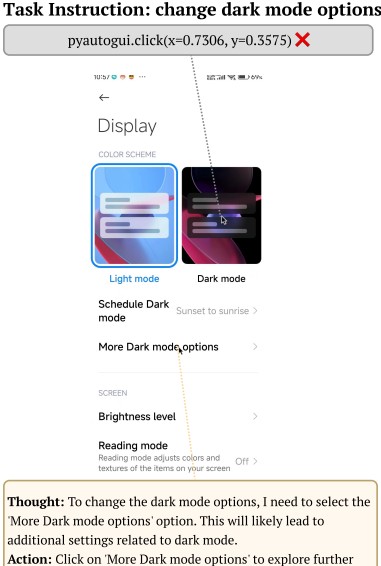

**Thought:** To change the dark mode options, I need to select the 'More Dark mode options' option. This will likely lead to additional settings related to dark mode.
**Action:** Click on 'More Dark mode options' to explore further settings.
pyautogui.click(x=0.4991, y=0.6742) ✅

Figure 6: Self-plan examples on different environments.

### E.3 Aguvis Trajectories Examples on Online Evaluation

### E.3.1 Mind2Web-Live Case: Aguvis-72B as Planner and Grounder

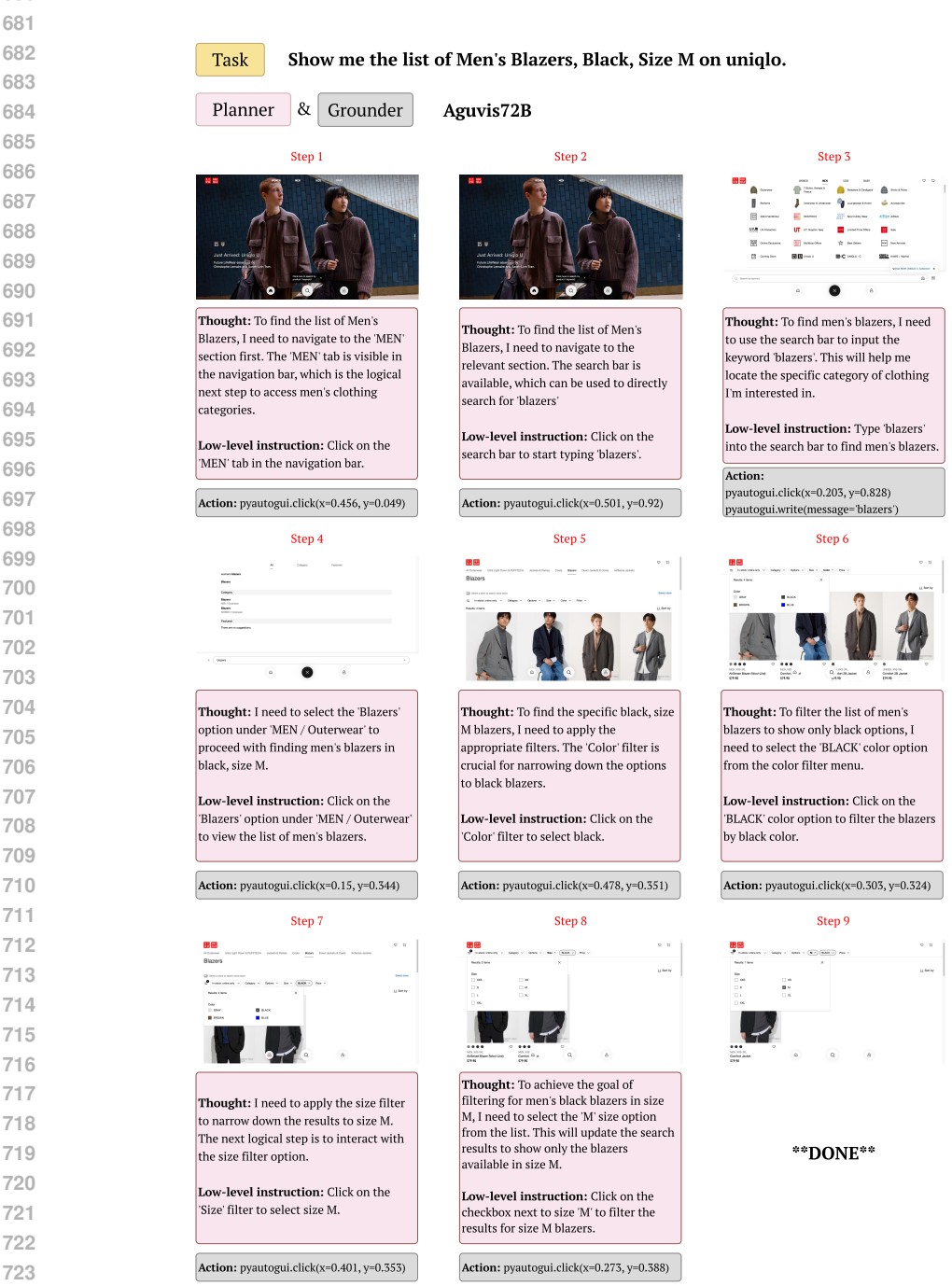

Figure 7: Example of Aguvis-72B as planner and grounder executing Mind2Web-Live task. Due to space limitations, we present here the trajectory generated guided by `Thought`.

### E.3.2  Mind2Web-Live Case: GPT-4o as Planner and Aguvis-7B as Grounder

**Task** Find a walkthrough guide for Assassin's Creed Valhalla on ign

**Planner** GPT-4o  **Grounder** Aguvis7B

**Step 1**

**Thought:** To find a walkthrough guide for Assassin's Creed Valhalla on IGN, I should use the search functionality. The search icon is visible on the left sidebar. Clicking on it should allow me to enter the search query.

**Low-level instruction:** click('search icon')

**Action:** pyautogui.click(x=0.033, y=0.4014)

**Step 2**

**Thought:** The search bar is now visible. To find a walkthrough guide for Assassin's Creed Valhalla, I need to enter the search query in the search bar.

**Low-level instruction:** fill('search input field', 'Assassin\'s Creed Valhalla walkthrough')

**Action:**
pyautogui.click(x=0.4991, y=0.3417)
pyautogui.write(text="Assassin's Creed Valhalla walkthrough")

**Step 3**

**Thought:** The search results for "Assassin's Creed Valhalla walkthrough" are displayed. The first result is "Assassin's Creed Valhalla." This is likely to contain the walkthrough guide we are looking for. I should click on this result to proceed.

**Low-level instruction:** click('Assassin\'s Creed Valhalla')

**Action:** pyautogui.click(x=0.4991, y=0.4847)

**Step 4**

**Thought:** To find a walkthrough guide, I should look for relevant sections or links on this page. The page has tabs like Overview, Playlists, and Reviews. The walkthrough guide is likely to be under one of these tabs, possibly the Overview or Playlists tab.

**Low-level instruction:** click('Overview tab')

**Action:** pyautogui.click(x=0.2115, y=0.7222)

**Step 5**

**Thought:** I am currently on the Assassin's Creed Valhalla page on IGN, and I have clicked on the "Overview" tab. To find the walkthrough guide, I should look for a section or link that specifically mentions "walkthrough" or "guide."

**Low-level instruction:** scroll(0, 300)

**Action:** /

**Step 6**

**Thought:** To find the walkthrough guide, I should look for a section or link that specifically mentions guides or walkthroughs. Since I don't see any relevant links or sections immediately, I should scroll further down the page to see if more content is revealed.

**Low-level instruction:** scroll(0, 500)

**Action:** /

**Step 7**

**Thought:** There is also a "Walkthrough" link under the "Top Guide Sections" on the right side. Clicking the "Walkthrough" link should take me directly to the walkthrough guide.

**Low-level instruction:** click('Walkthrough')

**Action:** pyautogui.click(x=0.5852, y=0.7681)

**Step 8**

**\*\*DONE\*\***

Figure 8: Example of GPT-4o as planner and Aguvis-7B as grounder executing Mind2Web-Live task.

### E.3.3 ANDROIDWORLD CASE: AGUVIS-72B AS PLANNER AND GROUNDER

**Task** — **Delete all but one of any expenses in pro expense that are exact duplicates, ensuring at least one instance of each unique expense remains.**

**Planner** & **Grounder** **Aguvis72B**

Step 1

**Thought:** To achieve the goal of managing expenses in Pro Expense, I need to open the app first. The list of apps includes Pro Expense, which is relevant to the task at hand.

**Low-level instruction:** Open the Pro Expense app using mobile.open_app.

**Action:** mobile.open_app(app_name='Pro Expense')

Step 2

**Thought:** To achieve the goal of deleting duplicate expenses, I need to access the list of all expenses. The current screen shows recent expenses, but I need to navigate to a section where I can view and manage all expenses. The menu icon at the top left is likely the gateway to accessing more options, including the full list of expenses.

**Low-level instruction:** Click on the menu icon at the top left to access more options.

**Action:** pyautogui.click(x=0.501, y=0.92)

Step 3

**Thought:** To achieve the goal of deleting duplicate expenses, I need to access the Expense Logs to view all expenses and identify duplicates. The Expense Logs option is visible in the menu.

**Low-level instruction:** Click on 'Expense Logs' to view the list of expenses.

**Action:** pyautogui.click(x=0.272, y=0.317)

Step 4

**Thought:** I need to delete one of the duplicate 'Public Transit' entries to ensure only one unique entry remains. This aligns with the goal of removing duplicate expenses.

**Low-level instruction:** Click on one of the 'Public Transit' entries to delete it.

**Action:** pyautogui.click(x=0.545, y=0.614)

Step 5

**Thought:** I need to delete duplicate entries, ensuring at least one unique entry remains. The current focus is on the 'Public Transit' entry, which seems to be a duplicate. I should delete this entry to meet the goal.

**Low-level instruction:** Click on the delete icon next to the 'Public Transit' entry to remove it.

**Action:** pyautogui.click(x=0.909, y=0.693)

Step 6

**Thought:** I need to confirm the deletion of the duplicate 'Public Transit' entry to ensure only one unique entry remains.

**Low-level instruction:** Click on the 'CONFIRM' button to delete the selected duplicate entry.

**Action:** pyautogui.click(x=0.532, y=0.923)

Step 7

**DONE**

Figure 9: Example of AGUVIS-72B as planner and grounder executing AndroidWorld task. Due to space limitations, we present here the trajectory generated guided by `Thought`.

### E.3.4 ANDROIDWORLD CASE: GPT-4O AS PLANNER AND AGUVIS-7B AS GROUNDER

Figure 10: Example of GPT-4o as planner and AGUVIS-7B as grounder executing AndroidWorld task.

### E.4 AGUVIS ON REAL-WORLD SENARIOS GENERALIZATION

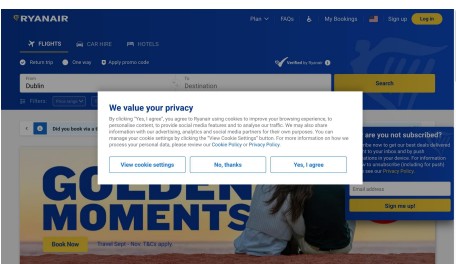 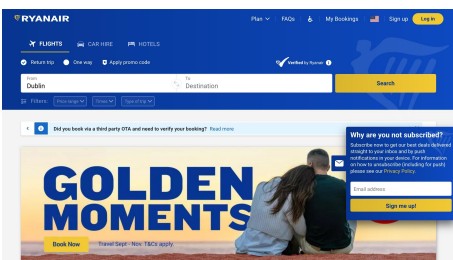

**Thought:** To proceed with finding a flight, I need to dismiss the privacy pop-up to access the search functionality.
**Low-level instruction:** Click 'No, thanks' to dismiss the privacy pop-up.
**Action:** pyautogui.click(x=0.498, y=0.603)

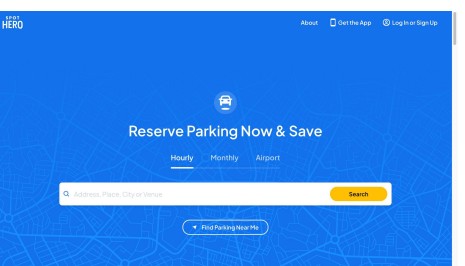 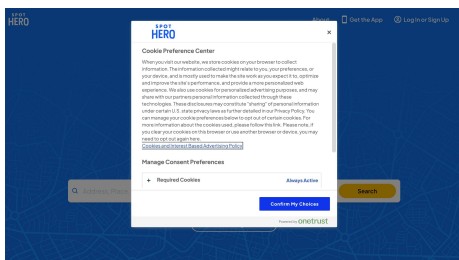

**Thought:** To proceed with contacting support, I need to close this overlay to access the main page where the 'Contact Us' or 'Support' link is likely located.
**Low-level instruction:** Click the 'X' button to close the cookie consent overlay.
**Action:** pyautogui.click(x=0.704, y=0.093)

Figure 11: Example of AGUVIS Closing Pop-ups in Real-world Scenarios

