# OpenReview forum: "Aguvis: Unified Pure Vision Agents for Autonomous GUI Interaction"
_ICLR.cc/2025/Conference — Submitted to ICLR 2025_

### Official Review · Reviewer_pn3z · 2024-10-28

**Soundness:** 3
**Presentation:** 2
**Contribution:** 2
**Rating:** 6
**Confidence:** 5

**Summary:**

This paper proposes Aguvis, a UI agent capable of understanding and interacting with multiple types of digital devices. The authors recognize the existing challenges, such as redundancy in textual representations and heterogeneity in action spaces across platforms, in UI agent research. To tackle these challenges, the authors organize existing datasets while unifying the agent's action space across mobile phones, web browsers, and desktop software. Besides, the authors introduce a two-stage training method to enhance the understanding and planning capabilities of the Aguvis agent. Comprehensive experiments are conducted to justify the authors' designs.

**Strengths:**

1. The authors conduct comprehensive experiments across multiple digital platforms.
2. The writing is neat and the paper is easy to follow.
3. The authors introduce a key design, i.e., integrating planning with action grounding, which overcomes the limitation of existing planning-then-grounding methods.

**Weaknesses:**

1. The reviewer doubts that the authors' method, Aguvis, effectively addresses two of the three challenges (i.e., Enhancing Pure Vision Framework, and Unification Across GUI Environments) in the Intro section.
1.1 To tackle the first challenge, the authors utilize an agent framework with pure vision input and curate multi-platform datasets to train the agent. However, the authors have not conducted ablation studies on the impact of introducing these datasets listed in Tables 9 and 10, thereby providing no practical insights into whether simply combining a pile of datasets can benefit all device domains. The authors are expected to isolate the datasets from different device domains to justify that it is unifying the observation, instead of simply increasing the data amount, that contributes to the final performance gains.
1.2 To unify the action space, the authors design a set of pyautogui functions as a union of the allowed actions across platforms. However, the authors still have not conducted ablation studies to justify this design. The authors are expected to conduct detailed ablation experiments to confirm that unifying action spaces is better than using the specialized action space of each device type.

2. The authors introduce several special designs to improve the agent's performances, but fail to provide solid experiments to justify them. For example, L248 states that 'This approach significantly accelerates training by maximizing the use of each image without compromising accuracy', but the experiment confirming this point is nowhere to be seen. Additionally, the expression 'we assume ...' in L250 renders the paper informal and unsolid. The authors cannot say this in an academic paper if no experiments justify that the GUI understanding capability is authentically  "robust".

3. The authors generate reasoning steps in the training data (L203). However, no experiments are conducted to prove the usefulness of this innovation.

4. The experiments that compare Aguvis with existing UI grounding models are not fair enough. In Table 1, the Aguvis finetuned on Qwen2-VL (a strong VLM pretrained with massive UI data) is compared with UGround and SeeClick, which are based on VLMs without being pretrained with massive UI data. This comparison is believed to be unfair and hard to demonstrate the superiority of Aguvis. The authors are suggested to organize the experiment more carefully.

5. No qualitative experiments, nor visualized examples are presented, making it hard for readers to understand the differences between Aguvis and existing methods.

**Questions:**

1. L152 states "Generally, the input length of accessibility tree observation is 6k tokens, and HTML is 4k tokens". Which data source is used to calculate these statistics?

2. Why do the authors not report the Step SR of CogAgent in Table 2?

3. What will the performance of Aguvis be on the UI planning benchmarks if Aguvis is trained with only stage 2?

4. Typos: Framwork (L51)

---

> ### Author Response · Authors · 2024-11-22
> **Official Comment by Authors (1/4)**
>
> Thank you for taking the time to review our work and providing constructive feedback! We greatly appreciate your recognition of the comprehensive experiments we conducted across multiple digital platforms, which demonstrate the the generalizability and promising performance of our approach. Additionally, we are pleased that you highlighted our integration of planning with action grounding, which is clearly key factor of advancing autonomous GUI agent. We are committed to fully open-sourcing our roadmap to foster future research.
>
> We also noticed you have some constructive questions about our work, and we're happy to elaborate further below!
>
> ---
>
> >**W1: Dataset ablation studies on impact of datasets from different device domains with unified action space.**
>
> A: We understand your concerns and would like to further clarify this concern:
>
> - We would like to respectfully emphasize that observation and action spaces are typically coupled. For example, with text-based observations, the corresponding click action would be click(element_id), while with pure vision observations, it would be click(x, y). Simply mixing non-unified data makes it difficult for the model to benefit mutually from different data sources, which is exactly the challenge we aim to address.
>
> - By utilizing pure vision inputs and a unified action space, we not only achieve mutual performance gains but also pave the way for developing more generalizable GUI agents. This approach eliminates the dependency on system-level APIs to access UI source code like HTML or Accessibility trees, relying solely on screenshots instead. Consequently, it enhances the agent's applicability across different platforms and devices.
>
> - However, we understand your concerns about dataset ablation. We have added new experiments to verify the impact of cross-platform data on downstream tasks. We trained models using the Mind2Web training set, expanded web-only data, and the full Stage 2 data including both web and mobile, then tested their performance on web tasks. The results are as follows:
>
> | Training Data| Cross-Task | Cross-Website | Cross-Domain |
> |---|---|---|---|
> | Web + Mobile (Stage 2 Equivalent) | 58.5 | 55.4 | 54.8 |
> | Web Only | 53.1 | 50.3 | 52.2 |
> | Mind2Web Only | 50.9 | 44.9 | 47.7 |
>
> Our results indicate that expanding the web domain data enhances performance on web tasks. Notably, incorporating mobile data leads to additional improvements in web domain performance, as evidenced by the higher performance when using the combined Web + Mobile dataset.
>
> These findings demonstrate that unified agent trajectory data can indeed achieve cross-platform benefits. The improved performance with the inclusion of mobile data validates our motivation to use a unified action space and pure vision inputs, as it enables the model to generalize better across different domains.
>
> ---
>
> >**W2: L248 states that 'This approach significantly accelerates training by maximizing the use of each image without compromising accuracy, but the experiment confirming this point is nowhere to be seen. Additionally, the expression "we assume" and "robust" are informal.**
>
> A:
> - In our preliminary experiments, we investigated the effect of the grounding packing strategy. We trained two models on the SeeClick web dataset, one employing the strategy and the other excluding it. We observed that grounding packing could significantly accelerate training efficiency, reducing overall training GPU hours from 6 to 1. Moreover, we found this strategy didn't hurt performance and even slightly improved the performance (76.8 vs. 73.3 on the ScreensSpot Web split). These results have been added to Appendix Section C.2.
>
> - Additionally, Thank you for pointing out our inappropriate wording. We have revised the term "assume" and "robust," replacing it with more scientifically rigorous language to ensure clarity and precision.
>
> ---

---

> ### Author Response · Authors · 2024-11-22
> **Official Comment by Authors (2/4)**
>
> >**W3: Justification reasoning step (inner monologue) generation**
>
> A: Thank you for your insightful feedback. The role of reasoning (inner monologue) is indeed significant. We think that the role of inner monologue can be attributed to two factors:
> - The generation of an inner monologue elicits the reasoning process concerning current actions.
> - The produced reasoning inner monologue provides informative context that informs subsequent planning stages.
>
> To further validate the importance of inner monologue, we conducted ablation studies to demonstrate the performance improvements achieved by incorporating inner monologue (IM) in training:
>
> | Setting | ScreenSpot | Multimodal-Mind2Web ||| AndroidControl ||
> |---|---|---|---|---|---|---|
> | | | Cross-Task | Cross-Website | Cross-Domain | High-Level | Low-Level |
> |AGUVIS|84.4|58.5|55.4|54.8|61.5|80.5|
> |AGUVIS w/o IM|79.3|55.4|53.7|54.9|60.3|69.1|
>
> Our findings indicate that:
> - Inner monologue clearly enhances performance on planning benchmarks.
> - Inner monologue can also improves low-level instruction grounding. This is because the low-level instructions in our augmented trajectory act as atomic instruction and grounding action pairs, enhancing GUI grounding ability.
>
> Overall, this ablation study clearly demonstrates the performance improvements brought by incorporating inner monologue in training, further justifying its effectiveness and supporting our analysis.
>
> ---
>
> >**W4: Due to the use of a more powerful pre-trained Qwen2-VL, it is difficult for current comparison to prove the superiority of AGUVIS.**
>
> A: Thank you for bringing up this important concern. We acknowledge that using a powerful backbone like Qwen2-VL could make it challenging to attribute performance gains solely to our AGUVIS methodology. To address this, we emphasize that AGUVIS is designed to be model-agnostic, enhancing GUI agent capabilities through two essential stages including grounding and planning\&reasoning.
>
> In our original submission (Section 4.2 and Table 8), we demonstrated that applying AGUVIS to a weaker backbone model like LLaVA still results in surpassing previous SOTA performances and achieves results comparable to those using Qwen2-VL.
>
> To further substantiate that the improvements stem from our methodology rather than the inherent strength of the backbone model, we conducted comprehensive ablation studies on both Qwen2-VL and LLaVA backbones. The results are summarized below:
>
> | $AGUVIS_{Qwen2VL}$ | ScreenSpot | Multimodal-Mind2Web ||| AndroidControl ||
> |---|---|---|---|---|---|---|
> | | | Cross-Task | Cross-Website | Cross-Domain | High-Level | Low-Level |
> |Stage 1 → 2|84.4|58.5|55.4|54.8|61.5|80.5|
> |w/o Stage 1|77.4|59.7|55.3|55.8|58.8|79.8|
> |w/o Stage 2|81.8|50.9|45.2|45.3|58.0|75.6|
> |w/o Stage 1 & 2|55.3|50.9|44.9|47.7|59.1|59.2|
>
>
>
> | $AGUVIS_{LLaVA}$  | ScreenSpot | Multimodal-Mind2Web ||| AndroidControl ||
> |---|---|---|---|---|---|---|
> | | | Cross-Task | Cross-Website | Cross-Domain | High-Level | Low-Level |
> |Stage 1 → 2|81.2|55.3|50.0|50.8|60.7|82.4|
> |w/o Stage 1|71.3|42.5|40.3|42.8|61.4|80.5|
> |w/o Stage 2|70.0|43.4|39.0|40.7|54.9|65.6|
> |w/o Stage 1 & 2|3.8|33.8|30.5|32.4|50.4|50.0|
>
> The ablation results reveal several key insights:
>
>
> 1. **Importance of Stage 1 (GUI Grounding)**: For ScreenSpot benchmark, both backbones show significant performance drops when Stage 1 is omitted, highlighting the necessity of diverse GUI grounding data provided in Stage 1 to handle the comprehensive evaluations of ScreenSpot.
>
> 2. **Impact of Stage 2 (Planning & Reasoning)**: For planning benchmarks like MM-Mind2Web & AndroidControl, excluding Stage 2 leads to notable declines in performance, especially on planning and reasoning tasks.
>
> 3. **Model-Agnostic Effectiveness**: Despite starting with lower baseline performance, LLaVA experiences substantial gains after applying the AGUVIS pipeline, surpassing previous SOTA methods and achieving results comparable to Qwen2-VL.
>
> These comprehensive ablation studies confirm that the superiority of AGUVIS arises from our innovative training methodology, rather than the inherent capabilities of the backbone model. By systematically enhancing both GUI grounding and planning abilities, AGUVIS delivers significant improvements across various benchmarks and backbones.

---

> > ### Author Response · Authors · 2024-11-22
> > **Official Comment by Authors (3/4)**
> >
> > ---
> >
> > >**W5: Qualitative experiments and visualized examples**
> >
> > A: Thank you for your constructive suggestion regarding the inclusion of qualitative experiments and visualized examples. In response to your feedback, we have expanded our appendix to include comprehensive analyses and visualizations. Specifically, we have added the following:
> > - Visualization of AGUVIS Data Collection Components (Appendix B.2)
> > - Training Example Data Schema and Prompt Template (Appendix C.1)
> > - Detailed Evaluation Prompts (Appendix D.3.1)
> > - Examples from Each Online Evaluation (Appendix E.3)
> > - Qualitative Human Study on Data Augmentation Pipeline Accuracy (Appendix B.3)
> >
> > We hope that these additions address your concerns and enrich the presentation of our work. Please let us know if there are specific aspects or additional examples you would like us to elaborate on.
> >
> > ---
> >
> > >**Q1: What data source was used to calculate token length statistics?**
> >
> > A: The 6k token length for the accessibility tree is based on OSWorld Figure 6, which indicates that a context length of 6000 tokens covers about 90% of cases for a single accessibility tree observation. The 4k token length for HTML is derived from the online evaluation Mind2Web-live in Table 4 of our paper, with the input length of HTML illustrated in Figure 2. We have cited the data source in the revised version. Thank you for pointing this out!
> >
> > ---
> > >**Q2: Why isn't CogAgent's Step SR reported in Table 2?**
> >
> > A: In Table 2 for Multimodal Mind2Web, we only report element accuracy for SeeClick and CogAgent. This is because the original SeeClick and CogAgent models were evaluated on Mind2Web, not Multimodal Mind2Web, making the examples misaligned and incomparable. Therefore, we referenced the results from UGround, where they report the element accuracy of the SeeClick and CogAgent models on Multimodal Mind2Web, striving to comprehensively present all previously representative methods. We have updated this explanation in Appendix D.2 of the revised version.

---

> > > ### Author Response · Authors · 2024-11-22
> > > **Official Comment by Authors (4/4)**
> > >
> > > ---
> > >
> > > > **Q3: What would AGUVIS's performance be if trained with only stage 2?**
> > >
> > > A: Based on original Table 6(Table 7 in revised version), we further added experiments that only train stage 2 (AGUVIS w/o Stage 1), with results shown in the table below.
> > >
> > > | Setting | ScreenSpot | Multimodal-Mind2Web ||| AndroidControl ||
> > > |---|---|---|---|---|---|---|
> > > | | | Cross-Task | Cross-Website | Cross-Domain | High-Level | Low-Level |
> > > |AGUVIS|84.4|58.5|55.4|54.8|61.5|80.5|
> > > |AGUVIS w/o Stage 1|77.4|59.7|55.3|55.8|58.8|79.8|
> > > |AGUVIS w/o Stage 2|81.8|50.9|45.2|45.3|58.0|75.6|
> > >
> > > Analysis of Results:
> > > 1. Impact on ScreenSpot (GUI Grounding Performance):
> > >     - The performance on the ScreenSpot benchmark drops from 84.4% to 77.4% when Stage 1 is omitted.
> > >     - ScreenSpot evaluates GUI grounding across a diverse set of images and domains, including web, desktop, and mobile interfaces. Stage 1 provides extensive and diverse GUI grounding training data, which is crucial for high performance on this benchmark. Without Stage 1, the model lacks the necessary exposure to varied GUI elements, leading to reduced grounding capabilities.
> > > 2. Impact on MM-Mind2Web and AndroidControl (Planning and Reasoning Tasks):
> > >     - Performance maintained without stage 1.
> > >     - Potiential two reasons:
> > >         - Potiential Pre-trained capabilities of Qwen2-VL: The backbone model Qwen2-VL was pre-trained on natural image grounding tasks, which provides it with inherent grounding abilities even without Stage 1 GUI grounding training.
> > >         - Extensive Stage 2 Trajectory Data: Stage 2 involves a large amount of trajectory data including grounding action pairs. This extensive training enables the model to effectively handle grounding in Mind2Web and AndroidControl, even in the absence of Stage 1.
> > >
> > > To further support our analysis, we refer to the ablation study conducted on the LLaVA model (as presented in Table 8 of our paper). The results are as follows:
> > >
> > > | Setting | ScreenSpot | Multimodal-Mind2Web |||
> > > |---|---|---|---|---|
> > > | | | Cross-Task | Cross-Website | Cross-Domain |
> > > |**AGUVIS-LLaVA**|81.2|55.3|50|50.8|
> > > |**AGUVIS-LLaVA w/o Stage 1**|71.3|42.5|40.3|42.8|
> > > |**AGUVIS-LLaVA w/o Stage 2**|70|43.4|39.0|40.7|
> > >
> > > We found that LLaVA model shows significant performance drops on both ScreenSpot and MM-Mind2Web when either Stage 1 or Stage 2 is omitted. Based on these results with two backbones, we can conclude that:
> > > - Qwen2-VL: Due to its pre-training on natural image grounding, it can maintain reasonable performance on planning tasks without Stage 1.
> > > - LLaVA: Lacks such pre-training, thus both Stage 1 and Stage 2 are critical for achieving high performance.
> > >
> > >
> > > Additionally, the consistent improvements across both Qwen2-VL and LLaVA backbones demonstrate that our AGUVIS methodology is effective regardless of the underlying model, which highlights the universality and adaptability of our training approach.
> > >
> > > These findings underscore the importance of incorporating both Stage 1 and Stage 2 in the training pipeline to achieve optimal performance across diverse GUI tasks and benchmarks. By providing comprehensive GUI grounding and enhancing planning and reasoning abilities, our AGUVIS approach ensures that models can generalize effectively to various domains and platforms.
> > >
> > > ---
> > >
> > > > **Q4: L51 Typo**
> > >
> > > A: Thank you for noticing this! We've corrected it in the updated version.
> > >
> > > ---
> > > We sincerely appreciate your detailed feedback. We hope the above response can address all your concerns. If you have any questions, we are pleased to provide further clarification!

---

> > > > ### Comment · Reviewer_pn3z · 2024-11-26
> > > > **Further questions**
> > > >
> > > > Thank you the authors for providing the detailed experiments and analysis. My concerns have been almost addressed, but several points are still unclear:
> > > >
> > > > 1. Why do the authors take a form of pyautogui during stage 1 training, which is solely for cultivating GUI grounding capability? What if just use plain coordinate outputs in stage 1?
> > > >
> > > > 2. Why do the authors not place the action space definition in the prompt during training and inference? Will this undermine evaluation performances?

---

> > > > > ### Author Response · Authors · 2024-11-26
> > > > >
> > > > > Thank you for your follow-up questions! We're so glad to hear that most of your concerns have been addressed! We are also pleased to further explain our action space design:
> > > > >
> > > > > ---
> > > > > > **Why do the authors take a form of pyautogui during stage 1 training, which is solely for cultivating GUI grounding capability? What if just use plain coordinate outputs in stage 1?**
> > > > >
> > > > > A: Thank you for noticing this design choice! We use PyAutoGUI actions during both stage 1 and stage 2 training to teach the model not only how to localize elements with coordinates but also to inject **GUI control capabilities**. This approach naturally integrates coordinate grounding with action-taking.
> > > > >
> > > > > During the **template-based grounding data augmentation** for stage 1, we employ diverse templates to combine coordinates with actions. This provides training examples for:
> > > > > - **Atomic actions** such as "click" (e.g., `pyautogui.click(0.6, 0.8)`), and
> > > > > - **Primitive skills** such as drag-to-select (e.g., `pyautogui.moveTo(0.1, 0.3)` followed by `pyautogui.dragTo(0.6, 0.8)`).
> > > > >
> > > > > This design allows AGUVIS-G to effectively learn PyAutoGUI as a unified action space during stage 1, embedding both grounding and action-taking capabilities. This early integration enables the model to seamlessly transfer these **primitive GUI control skills** to stage 2 planning&reasoning training, improving learning efficiency for more complex interactions and tasks.
> > > > >
> > > > > ---
> > > > >
> > > > > > **Why do the authors not place the action space definition in the prompt during training and inference? Will this undermine evaluation performances?**
> > > > >
> > > > > A: While training AGUVIS with action definitions included in the prompts is technically possible, we chose to exclude these definitions for practical reasons. Including the definitions of [all atomic PyAutoGUI actions](https://pyautogui.readthedocs.io/en/latest/quickstart.html) would add approximately 1,000 tokens as a prompt prefix. This would significantly increase the computational cost for both training and inference, reducing overall efficiency.
> > > > >
> > > > > Thanks to the **template-based augmentation** in stage 1, AGUVIS efficiently learns these GUI control actions internally, without requiring explicit action definitions during inference. This stage 1 training enables the model to generalize and transfer its action capabilities seamlessly to stage 2, avoiding any loss in evaluation performance.
> > > > >
> > > > > Moreover, to maintain flexibility and adaptability across different environments, we have incorporated a **pluggable action space design**. This allows AGUVIS to extend its action space for specific benchmarks that require additional functionality. For example, in the AndroidWorld mobile environment (see Appendix A.2), AGUVIS leverages provided Android system-level functions such as `mobile.open_app`, `mobile.home`, and `mobile.back`. These high-level actions enable AGUVIS to perform better in benchmarks with unique requirements.
> > > > >
> > > > > This approach strikes a balance between efficiency, flexibility, and performance, ensuring that AGUVIS can adapt to diverse tasks and environments while maintaining consistent GUI control capability.

---

> > > > > > ### Comment · Reviewer_pn3z · 2024-11-27
> > > > > > **Further Questions**
> > > > > >
> > > > > > Thanks for the detailed response! Further questions about the implementation of the **template-based grounding data augmentation** arise:
> > > > > >
> > > > > > 1. Why do the authors use the drag-to-select as the response format for grounding tasks? Is it used for tasks requiring bounding box outputs?
> > > > > >
> > > > > > 2. Do the authors also reformat the grounding referring expression as an intent format, such as "click/moveTo/drag <element referrign expression>"?

---

> > > > > > > ### Author Response · Authors · 2024-11-27
> > > > > > >
> > > > > > > Thank you for your prompt feedback! We greatly appreciate your time and attention in discussing our work. We are pleased to provide more details about our template-based grounding data augmentation design. We would like to address two questions together, hoping this will help clarify our design for you!
> > > > > > >
> > > > > > > ---
> > > > > > >
> > > > > > > During stage 1, the original data comprises pairs of referring expressions and their corresponding bounding box coordinates $(x_1, y_1, x_2, y_2)$ derived from GUI screenshots. Examples include:
> > > > > > >
> > > > > > > - **Textual elements:** ("More Information", [0.1, 0.3, 0.6, 0.8])
> > > > > > > - **Iconic elements:** ("Share Icon", [0.1, 0.3, 0.2, 0.4])
> > > > > > >
> > > > > > > Previous methods typically use these referring expressions as instructions/intents and the bounding boxes as prediction targets.
> > > > > > >
> > > > > > > To enhance this dataset, our template-based data augmentation strategy **transforms and reformats** these referring expressions and bounding box pairs into **diverse instruction-action mappings**. This step prepares the model for a wide array of GUI control tasks.
> > > > > > >
> > > > > > > 1. **Atomic Action**
> > > > > > >    We generate diverse templates for straightforward GUI actions such as `click`, `doubleClick`, `rightClick`, and `moveTo`. **These templates directly map grounding referring expressions to PyAutoGUI-compatible action commands.** For example:
> > > > > > >
> > > > > > >    - Original Data: ("Share Icon", [0.1, 0.3, 0.2, 0.4])
> > > > > > >    - Augmented Instruction: `Click Share Icon`
> > > > > > >    - Corresponding Action: `pyautogui.click(0.15, 0.35)`
> > > > > > >
> > > > > > >    By varying templates and refining bounding box coordinates, we ensure broader coverage of PyAutoGUI functions.
> > > > > > >
> > > > > > > 2. **Primitive Skill**
> > > > > > >    To equip the model with foundational skills for complex tasks, **we augment the dataset with instructions requiring multiple actions, such as dragging, highlighting, or copying.** These skills are crucial for completing compound tasks effectively. For example, given the original text element: ("Price: $100", [0.1, 0.3, 0.6, 0.8]), we can use PyAutoGUI actions to first move to the center of the left edge and then drag to the center of the right edge to select the text. Here's how it can be done:
> > > > > > >
> > > > > > >    - **Highlighting Text:**
> > > > > > >      ```
> > > > > > >      User: Drag to select “Price: $100”
> > > > > > >      Agent:
> > > > > > >      pyautogui.moveTo(0.1, 0.55)
> > > > > > >      pyautogui.dragTo(0.6, 0.55)
> > > > > > >      ```
> > > > > > >
> > > > > > >    - **Copying Text:**
> > > > > > >      ```
> > > > > > >      User: Copy “Price: $100” as answer.
> > > > > > >      Agent:
> > > > > > >      pyautogui.moveTo(0.1, 0.55)
> > > > > > >      pyautogui.dragTo(0.6, 0.55)
> > > > > > >      pyautogui.hotkey('ctrl', 'c')
> > > > > > >      ```
> > > > > > >
> > > > > > >    These scenarios ensure that the model internalizes common action patterns in stage 1, building a foundation for more intricate tasks in stage 2 trajectory training.
> > > > > > >
> > > > > > > ---
> > > > > > >
> > > > > > > We hope this elaboration clarifies our template-based grounding data augmentation approach and demonstrates its importance in our training paradigm. We will update these details into our appendix. Please feel free to share additional questions or suggestions!

---

> > > > > > > > ### Comment · Reviewer_pn3z · 2024-11-27
> > > > > > > > **Final Comment**
> > > > > > > >
> > > > > > > > Thanks for providing the details of the proposed template-based grounding data augmentation!
> > > > > > > >
> > > > > > > > No more concerns and the rating has been raised to 6.

---

> > > > > > > > > ### Author Response · Authors · 2024-11-27
> > > > > > > > > **Thank you so much for raising our rating!**
> > > > > > > > >
> > > > > > > > > We sincerely appreciate your time and attention in discussing with us. Thank you for raising our rating; it truly encourages us to continue improving our work and exploring this exciting direction. We are more than happy to address any further questions you may have!

---

> ### Author Response · Authors · 2024-11-25
> **Follow-up Comment by the Authors**
>
> Dear Reviewer pn3z:
>
> We sincerely thank you for these constructive comments and evaluation of our paper. With the ICLR public discussion phase ending **in two days**, we kindly ask you to take a look at our responses. Our rebuttal also provided additional experiments in response to your concerns. Please let us know whether our response addresses your concerns or whether there is any further detail we can provide to help address these concerns.
>
> Thank you again for dedicating your time to reviewing our paper.

---

### Official Review · Reviewer_Vo6b · 2024-11-02

**Soundness:** 3
**Presentation:** 3
**Contribution:** 2
**Rating:** 5
**Confidence:** 5

**Summary:**

This paper presents AGUVIS, a vision-based framework for developing GUI agents. The authors compile datasets from various sources to cover multiple platforms, standardize them with a unified action space, and enhance them with intermediate planning steps and actions. They leverage a well-chosen vision-language model (VLM) within a two-stage training paradigm to create a visual-based GUI agent. The framework is evaluated on both offline and online GUI benchmarks, demonstrating promising results.

**Strengths:**

1. Performance: AGUVIS achieves strong performance across multiple benchmarks, including ScreenSpot, Multimodal Mind2Web, and Android Control, highlighting its versatility in visual-based agent applications.
2. Methodological Contribution: The paper offers a comprehensive roadmap for building robust visual-based agents, focusing on data curation and model training strategies.
3. Clarity: The paper is well-written and straightforward to follow, making complex concepts accessible to readers.

**Weaknesses:**

**1. Limited Novelty:** The main contribution of this paper is to pre-train VLMs with massive data combinations, but lack insights and are computationally intensive. Although AGUVIS successfully integrates existing datasets for vision-based GUI model training, it neither introduces new datasets nor proposes a novel module design.
Insufficient Detail in Key Areas:

**2. Data Curation (Augvis Collection)**
- Data collection and curation are crucial to this work, yet important details are omitted. For example, what specific VLM is used for generating inner monologue components? How is the accuracy of the generated observation descriptions, thoughts, and low-level action instructions ensured?
- Training and Implementation Details: The two-stage training procedure lacks sufficient description. Key aspects, such as the training schedule for each stage, are unclear. Additionally, in Stage 2, it would be helpful to explain how inputs like observations, thoughts, action histories, and observation histories are organized. Are there truncation strategies to handle long sequences? Supplementary figures illustrating these aspects would enhance clarity.

**3. Lack of Analysis** Further analysis would improve the clarity and depth of the paper:
- Unexpected Model Behavior on ScreenSpot: The self-planning model’s outperformance over the original instructions version on ScreenSpot is surprising, given that the ScreenSpot task involves simple grounding queries. Does this imply overfitting to Stage 2 training patterns? How does the model perform after Stage 1 training alone? Providing examples of the model’s planning outputs could deepen this analysis.
- Justification for Two-Stage Training: There is no experimental evidence supporting the decision to employ a two-stage training paradigm. Why not combine both stages into a single training pipeline?
- Effectiveness of Inner Monologue: The paper lacks evidence demonstrating the contribution of the VLM-generated inner monologue component to overall performance. The observed degradation in performance without Stage 2 training could simply result from training on more data rather than specifically leveraging inner monologue components.

**Questions:**

- Could the authors clarify the specific VLM used for generating inner monologue components? What measures ensure the accuracy of generated observations, thoughts, and action instructions?

- For Stage 1 training, does the grounding-packing strategy employ a causal attention pattern for the packed sequence, or is there a customized attention mask to prevent attention between different grounding samples?

---

> ### Author Response · Authors · 2024-11-22
> **Official Comment by Authors (1/3)**
>
> Thank you for taking the time to review our work and provide detailed feedback! We sincerely appreciate your recognition of our work as a comprehensive roadmap for developing a pure vision GUI agent, particularly in terms of data curation and training strategies. Constructing the entire pipeline for this framework is indeed challenging, as the pure vision setting is a novel and promising area in this field. Previous data and evaluation methods needed significant adaptation for this context. The challenge is heightened by our model's ability to plan independently during online realistic evaluations, rather than relying on closed-source models like GPT-4o. This makes AGUVIS an autonomous pure vision agent model. We are committed to fully open-sourcing our roadmap to support future research in this field.
>
> We also noted that you have concerns about some details of our work and we are pleased to explain further:
>
> ---
>
> >**W1: Limited novelty - mainly pre-training VLMs with data combinations**
>
> A: We sincerely thank the reviewer for their constructive feedback. Below, we address each concern regarding the novelty of our work:
>
> 1. Insights Provided:
>     - In Section 4, we presented extensive training ablations, demonstrated the model-agnostic capabilities of our approach with LLaVA, highlighted the efficiency advantages of pure vision modeling, and provided error analyses that emphasize the benefits of improved planning.
>     - To further address your concerns, we have added Appendix E, which includes additional training ablations on both Qwen2-VL and LLaVA models, analyses of dataset synergy, visualization examples from online evaluations, and experiments demonstrating generalization to real-world environments. These additions offer valuable insights into the viability and advantages of pure vision GUI agents, and we believe they will guide future research in this area.
> 2. Addressing Computational Intensity:
>     - Exploring Model Scalability: GUI tasks require complex decision-making. Therefore, it's important to explore the potential of high-capacity model. It worth to conduct evaluations across model scales (7B-72B) which can provide comprehensive insights into how model size impacts performance.
>     - Efficiency Enhancements: Our pure vision modeling approach naturally reduces token consumption compared to previous methods. To further mitigate computational demands, we have implemented packing techniques to accelerate training, as detailed in Appendix C.2. These strategies enhance efficiency without compromising performance.
> 3. Dataset Integration as a Novel Contribution:
>     - Novel Integration Pipeline: Integrating trajectory data with vision-based datasets was a significant challenge. We developed a novel data pipeline that ensures unified action space modeling and incorporates VLM augmentation with inner monologues, which are critical for task performance, as validated in our experiments.
>     - We created an agent data pipeline that effectively leverages existing datasets and augments them with additional inner monologues. In the future, it will incorporate more agent trajectory datasets. We committed to fully open-source this data pipeline as a resource for scalable agent research.
> 4. Novel Module Design:
>     - Pure Vision Agent Model: Our pure vision agent represents a novel approach to GUI agent modeling by unifying planning and grounding within a single framework. This design overcomes previous scalability and dependency issues, enabling broader applicability without relying on closed-source models or specific environments.
>
> We believe that our work presents significant novel contributions both in methodology and practical applications. We hope that these clarifications address your concerns regarding the novelty of our work. We are happy to provide further details or engage in additional discussions to showcase the impact of our contributions!
>
> ---

---

> > ### Author Response · Authors · 2024-11-22
> > **Official Comment by Authors (2/3)**
> >
> > ---
> >
> > >**W2.1: Which VLM is used for generating inner monologue and how is the accuracy?**
> >
> > A:
> > - **During our data curation, we use GPT-4o to augment agent trajectory by adding explicit reasoning thought and low-level instruction.** We present the input prompt (shown in Appendix B.2) to GPT-4o for generating detailed, low-level instructions. Specifically, the prompt is consist of high-level goal, previous action history, the current ground truth action command, and the current observation image, where the target object is highlighted with a red box. We then ask GPT-4o to complete the reasoning process and provide a response in JSON format for easy parsing.
> > - **Our qualitative evaluation shows high accuracy of the augmented data.** Through a human study of samples from each datasets, 86.7% of the augmented data successfully demonstrated intermediate reasoning that aligned with both the ground truth actions and overall goal intention. Only 7.8% of cases were affected by dataset noise, and 5.5% showed misinterpretation of action intention in clean data.
> > - More detailed examples can be found in Appendix B.3 and Figure 4 of the revised paper.
> > ---
> >
> > >**W2.2: Training and Implementation Details**
> >
> > A: Thanks for your feedback on the implementation details. We share more details to solve your concerns below:
> > - **Training schedule**: We add more training details in Appendix C.
> > - **Input organization**: We have visualized the examples of our data collection in Appendix Section B.3 and our training example schema in Appendix C.
> > - **Sequence truncation strategies**: The maximum sequence length of tokens is set to 8192 for all models. This meets the need of our training because a 720p-resolution image takes 1196 tokens. Generally, this length will not surpass 8k context even with the informative context as shown in Appendix C.1.
> >
> >
> > ---
> >
> > >**W3.1: Surprising self-planning model behavior on ScreenSpot**
> >
> > A:
> >
> > - Thank you for your detailed feedback. We would like to clarify that not all queries from ScreenSpot are simple and straightforward. Many queries require the model to understand semantics and perform additional reasoning. To address your concerns, we have visualized these semantic result examples from both the stage 1 and stage 2 models in Appendix E.2.2.
> >
> > - We found that the stage 1 model effectively handles straightforward examples, such as clicking on a text element in an image. However, for more complex semantic queries shown in Appendix E.2.2, the model needs to incorporate additional knowledge and reason. High-quality stage 2 training can enhance the model's performance in these areas, underscoring the importance of Stage 2.
> >
> > ---
> >
> > >**W3.2: Justification for Two-Stage Training**
> >
> > A: To explore the impact of two-stage training versus joint training of stages 1 and 2, we conducted a controlled experiment. The results are presented in the table below:
> >
> > | Setting | ScreenSpot | Multimodal-Mind2Web ||| AndroidControl ||
> > |---|---|---|---|---|---|---|
> > | | | Cross-Task | Cross-Website | Cross-Domain | High-Level | Low-Level |
> > |AGUVIS (staged)|84.4|58.5|55.4|54.8|61.5|80.5|
> > |AGUVIS (joint)|85.0|56.1|53.1|55.6|59.2|80.9|
> >
> > - From the results, we observe that the overall performance differences between the two training setups are not significant. However, a clear trend emerges: the joint training setup (stages 1 and 2 trained together) enhances the model's performance on GUI grounding tasks such as ScreenSpot and the Low-Level tasks in AndroidControl. Conversely, it performs worse on tasks requiring planning, as indicated by the lower scores in the Multimodal-Mind2Web evaluations.
> > - This trend can be explained by the data composition in each stage. Stage 1 contains a larger volume of grounding data, which, when combined with stage 2 in joint training, dominates the optimization process. This dominance biases the model towards grounding capabilities at the expense of planning abilities. In contrast, stage 2 offers higher-quality data that is more aligned with the agent's deployment scenarios, especially concerning planning and complex grounding. This is why we opted to place stage 2 after stage 1 in a two-stage training process.
> >
> > - Recent research[1] suggests that introducing higher-quality data during the learning rate decay phase can enhance model performance. Inspired by this, a potential future training strategy could involve jointly training on data from both stages during the initial training phase. Then, during the learning rate decay phase, we would fine-tune the model using only the high-quality trajectory data from stage 2 to further refine its capabilities. We plan to explore this and other training methods in future work to optimize the balance between grounding and planning skills.
> >
> >
> > References:
> > [1] MiniCPM: Unveiling the Potential of Small Language Models with Scalable Training Strategies. Hu et al., 2024.
> >
> > ---

---

> > > ### Author Response · Authors · 2024-11-22
> > > **Official Comment by Authors (3/3)**
> > >
> > > >**W3.3: Evidence of inner monologue effectiveness**
> > >
> > > A: Thank you for your insightful feedback. We believe that the inner monologue plays a crucial role due to two main factors:
> > > - The generation of an inner monologue elicits the reasoning process concerning current actions.
> > > - The produced reasoning inner monologue provides informative context that informs subsequent planning stages.
> > >
> > > To validate the importance of the inner monologue (IM) and low-level instructions, we present the following ablation results:
> > >
> > > | Setting | ScreenSpot | Multimodal-Mind2Web ||| AndroidControl ||
> > > |---|---|---|---|---|---|---|
> > > | | | Cross-Task | Cross-Website | Cross-Domain | High-Level | Low-Level |
> > > |AGUVIS|84.4|58.5|55.4|54.8|61.5|80.5|
> > > |AGUVIS w/o IM|79.3|55.4|53.7|54.9|60.3|69.1|
> > >
> > > Our findings indicate that:
> > > - Inner monologue clearly enhances performance on planning benchmarks.
> > > - Inner monologue can also improves low-level instruction grounding. This is because the low-level instructions in our augmented trajectory act as atomic instruction and grounding action pairs, enhancing GUI grounding ability.
> > >
> > > Overall, this ablation study clearly demonstrates the performance improvements brought by incorporating the inner monologue in training, further supporting our analysis.
> > >
> > > ---
> > >
> > > >**Q1: Which specific VLM generated inner monologue components and how is accuracy?**
> > >
> > > A: Please kindly refer to W2.1 and Appendix Section B.3
> > >
> > > ---
> > >
> > > >**Q2: Does stage 1 training use causal attention or custom attention mask for packed sequences?**
> > >
> > > A:
> > > - In Stage 1, to expedite training, we concatenate multiple grounding pairs from the same image into a single example. This approach substantially enhances training efficiency without compromising performance. We didn’t modify the attention mask; instead, we employed a straightforward causal attention mechanism.
> > > - We conducted an ablation study using web grounding data and evaluated on the ScreenSpot web split. Our findings demonstrated that grounding packing significantly accelerated training efficiency, reducing the overall GPU hours required from 6 hours to 1 hour. Moreover, this strategy didn’t affect performance and even marginally outperformed the baseline model (76.8 vs. 73.3 on the ScreensSpot web split).
> > > - However, the custom attention mask you mentioned is definitely a method worth trying. Although implementing this with flash attention is relatively difficult, with the recently released flex attention, it should be more feasible. We will try to add this training optimization in our released code. Thank you for your suggestion!
> > >
> > > ----
> > >
> > > We sincerely appreciate your detailed feedback. We hope the above response can address all your concerns. If you have any questions, we are pleased to provide further clarification!
> > >
> > > ----

---

> > > > ### Author Response · Authors · 2024-11-25
> > > > **Follow-up Comment by the Authors**
> > > >
> > > > Dear Reviewer Vo6b
> > > >
> > > > We sincerely appreciate your time and effort in reviewing our paper. With the ICLR public discussion phase ending in two days, we would like to ensure our previous response addressed your concerns. If you have any further questions or additional feedback, we would be more than happy to address them. Thank you!
> > > >
> > > > Authors

---

> ### Author Response · Authors · 2024-11-26
>
> Dear Reviewer Vo6b,
>
> Hello! We appreciate your suggestions. Since the discussion period has started, we kindly ask you to take a look at our responses. Please let us know whether our response addresses your concerns or whether there is any further detail we can provide to help address them. We appreciate your time and consideration!
>
> Authors

---

> > ### Comment · Reviewer_Vo6b · 2024-11-28
> > **Post-response by Reviewer Vo6b**
> >
> > First of all, I sincerely thank the reviewer for the thorough and detailed feedback. I understand the effort it must have taken to provide such comprehensive insights and conduct additional experiments within a short timeframe.
> >
> > I would like to increase my rating from the original 3 to 5, as the response have addressed most of my concerns.
> >
> > However, my primary concern remains with the focus of this work.
> > While it emphasizes large-scale pretraining and various engineering optimizations, it still lacks significant technical innovation regarding the vision perception or language agents.

---

> ### Comment · Reviewer_Vo6b · 2024-11-28
> **Followup-questions**
>
> I want to thank for the author effort for including Appendix and include many information.
>
> Efficiency Enhancements: Our pure vision modeling approach naturally reduces token consumption compared to previous methods. To further mitigate computational demands, we have implemented packing techniques to accelerate training, as detailed in Appendix C.2. '''
> This is still unclear to me (regarding token consumption), could the author provide more explanation on this part?
>
> Novel Integration Pipeline: Integrating trajectory data with vision-based datasets was a significant challenge. We developed a novel data pipeline that ensures unified action space modeling and incorporates VLM augmentation with inner monologues, which are critical for task performance, as validated in our experiments.
> '''
> How does the integration pipeline work and how it differs from previous approaches.

---

> ### Comment · Reviewer_Vo6b · 2024-11-28
> **Follow-up Questions**
>
> **Regarding
> Novel Integration Pipeline**: Integrating trajectory data with vision-based datasets was a significant challenge. We developed a novel data pipeline that ensures unified action space modeling and incorporates VLM augmentation with inner monologues, which are critical for task performance, as validated in our experiments.
>
> Can the author provide more explanation how this integration pipeline works and how it differential or novel than previous methods such as GUI-Course?
>
> **Training cost**:
>
> Expensive training costs might be challenging for other scholars to afford.
>
> Could you provide details on the training costs for 7B/72B models at each stage? Are these models fully trained, or do they utilize LoRA for fine-tuning? Additionally, how does the performance compare across these methods? which will provide valuable reference for community. Thank you.

---

> ### Author Response · Authors · 2024-11-29
> **Follow Up Response (1/5)**
>
> We deeply appreciate you taking the time to respond to us and improve the rating. Your recognition of our comprehensive insights and additional experiments we conducted in a short period is truly encouraging for us to continue improving our work. We are very grateful for you taking the time to read our reply and raise more valuable questions. We are definitely more than happy to provide more explanations about these questions and discuss our contribution and future further.
>
> > **Follow Up Q1: This is still unclear to me (regarding token consumption), could the author provide more explanation on this part?**
>
>
> A: We are happy to provide further details regarding reducing token consumption, which we believe is one of the core advantages of pure vision GUI agent modeling. To control a GUI, most previous methods required access to the source code of the GUI, which on the web is represented by HTML, and on operating systems (like desktop and mobile OS) is represented by the accessibility tree. These text-based data structures represent the interactive elements in the interface, and the agent selects an element and performs an action to complete the task. **A clear drawback of this approach is that each observation is very long, and the length increases as the complexity of the GUI grows.** This results in large encoding costs for the agent model. For instance, even after extensive cleaning and pruning of these textual trees, the average token consumption for an accessibility tree on OS is around 6k tokens per observation, and 4k tokens for web HTML. These long observations lead to high overhead in both the training and inference phases.
>
> To address this, our pure vision approach uses screenshots of the interface as observations and controls the GUI through generated pyautogui actions and coordinates. For a 720p (1280*720) screenshot, thanks to dynamic resolution strategy of the NaViT image encoder, AGUVIS consumes only 1196 tokens while maintaining the resolution and aspect ratio of the screenshot. This significantly reduces token consumption, which improves training and inference efficiency. The efficiency advantage of this pure vision modeling, combined with its generalization as a unified representation of the GUI, further demonstrates the benefits of pure vision observation.
>
> You can also find more motivation explanation in our paper (`L51-59`, `L151-155`), as well as details about dynamic resolution strategy in the NaViT paper itself[1]. We also highlight the advantage of token consumption in Section 4.3 with Figure 2. If you'd like to know more about GUI observation and NaViT, we would be more than happy to provide further explanations!
>
> Reference
>
> [1] Patch n' Pack: NaViT, a Vision Transformer for any Aspect Ratio and Resolution. Dehghani et al., 2024.

---

> ### Author Response · Authors · 2024-11-29
> **Follow Up Response (2/5)**
>
> > **Follow Up Q2: Can the author provide more explanation how this integration pipeline works and how it differential or novel than previous methods such as GUI-Course?**
>
> A: Yes, we are pleased to provide further insights into our crucial data pipeline in building AGUVIS!
>
> The core motivation behind developing the AGUVIS data pipeline is to **unify the observation and action space** for effective data scaling, while **systematically incorporating data augmentation techniques** into both Stage 1 (GUI grounding) and Stage 2 (planning & reasoning). This unified pipeline fundamentally enables two key improvements in AGUVIS: **pure vision perception with a pluggable, unified action space** and **autonomous planning and reasoning** capabilities.
>
> ### Data Pipeline Overview
>
> The integration pipeline begins by standardizing both grounding and trajectory data across **different environments**. This step is crucial because existing datasets are often designed for distinct environments with mismatched observation and action spaces, making direct integration between them a challenge. Our approach extracts all observations as images and converts action annotations into a **unified coordinate-based grounding format**, which is compatible with **PyAutoGUI** for consistent action execution across different platforms. We illustrated the unified action space and an example of plugin action space for mobile as an example in Appendix A.1 and A.2.
>
> This standardization allows for significant and effective **data scaling**, expanding the grounding data to **1 million screenshot samples** and the trajectory data to **35K tasks** with **300K actions**. Compared to previous works, this represents an **order-of-magnitude increase** in dataset size and complexity. The unification of observation and action spaces also facilitates **cross-platform compatibility**, enabling AGUVIS to work seamlessly across various GUI environments. We have further demonstrated the effectiveness of cross-platform benefits in Table 16, where we demonstrate unified mobile trajectories can effectively improve the performance of web browsing.
>
> ### Novel Data Augmentation Strategies
>
> Once the data is unified, we introduce two novel **data augmentation strategies** designed to improve the model's performance in both **Stage 1 (GUI grounding)** and **Stage 2 (planning and reasoning)**:
>
> ------
>
> #### **Stage 1: GUI Grounding** Augmentation
>
> In Stage 1, our focus is on grounding the model’s perception of GUI elements and improving its basic GUI action capabilities. This involves using a **template-based data augmentation strategy** that transforms and reformats referring expressions and bounding box pairs into **diverse instruction-action mappings**. This step prepares the model for a wide range of GUI control tasks, enhancing its ability to generalize across different environments.
>
> 1. **Atomic Action Augmentation**
>     We generate a variety of templates for basic GUI actions, such as `click`, `doubleClick`, `rightClick`, and `moveTo`. These templates directly map grounding referring expressions (e.g., GUI element labels) to **PyAutoGUI-compatible actions**. For example:
>
>    - **Original Data:** ("Share Icon", [0.1, 0.3, 0.2, 0.4])
>    - **Augmented Instruction:** `Click Share Icon`
>    - **Corresponding Action:** `pyautogui.click(0.15, 0.35)`
>
>    By varying these templates and adjusting bounding box coordinates, we ensure that the model is exposed to a wide variety of basic GUI actions, thus broadening its coverage of PyAutoGUI functions.
>
> 2. **Primitive Skill Augmentation**
>     In addition to basic actions, we augment the dataset with instructions requiring **multiple actions** in sequence, such as **dragging**, **highlighting**, or **copying**. These skills are foundational for the model to complete more complex tasks. For example, given a text element ("Price: $100", [0.1, 0.3, 0.6, 0.8]), the model can perform the following actions:
>
>    - **Highlighting Text:**
>
>      ```
>      User: Drag to select "Price: $100"
>      Agent:
>      pyautogui.moveTo(0.1, 0.55)
>      pyautogui.dragTo(0.6, 0.55)
>      ```
>
>    - **Copying Text:**
>
>      ```
>      User: Copy "Price: $100" as answer.
>      Agent:
>      pyautogui.moveTo(0.1, 0.55)
>      pyautogui.dragTo(0.6, 0.55)
>      pyautogui.hotkey('ctrl', 'c')
>      ```
>
>    These scenarios enable the model to internalize common action patterns in Stage 1, establishing a solid foundation for more complex tasks in Stage 2.

---

> ### Author Response · Authors · 2024-11-29
> **Follow Up Response (3/5)**
>
> ------
>
> #### **Stage 2: Planning and Reasoning Augmentation**
>
> For Stage 2, which involves trajectory data and task planning, we introduce **GPT-4o-based inner monologue augmentation**. Specifically, we provide GPT-4o with a prompt that includes:
>
> - A high-level task goal
> - Previous action history
> - The current ground truth action
> - The current observation image, with the target object highlighted in a red box
>
> GPT-4o is then tasked with generating inner monologue including explicit reasoning and detailed low-level instructions. The output is returned in JSON format, facilitating easy parsing and integration into the model’s training process. We have shown the prompt in Appendix B.2 for your reference. We also illustrated our augmented trajectory data in Figure 4 and compared the difference with previous method in Figure 5.
>
> **Thanks to our unified data pipeline, we can effectively apply this pipeline on all trajectory data.** By incorporating **inner monologue** into the trajectory data, we significantly improve the model's ability to plan and reason about complex tasks, which is crucial for autonomous decision-making in real-world GUI environments. We have demonstrated the importance of inner monologue in W3.3 and Table 17.
>
> ------
>
> ### Key Differences from Existing Approaches
>
> We would also like to highlight our approach introducing several key advancements over previous methods such as GUICourse:
>
> 1. **Unified Observation and Action Space Design:** Previous methods like GUICourse usually operate on fragmented data spaces for web and mobile, which are often tailored to specific environments and requires fine-tune on difference downstream training data for platform/environment adaptation. Our unification data collection along with the use of PyAutoGUI for action commands, ensures that AGUVIS works seamlessly across different platforms and environments, a major advantage over previous systems.
> 2. **Effective Data Scaling for Cross-Platform:** Our pipeline supports a much larger data collection, expanding the grounding data to 1 million screenshot samples and trajectory data to 35K tasks with 300K actions—an order-of-magnitude increase in scale compared to prior works as shown in Table 10 & 11. We unify observation and action modalities, enabling broader cross-platform performance benefits as shown in Table 16.
> 3. **Dual-Stage Augmentation:** We integrate two complementary augmentation strategies—**template-based grounding augmentation**—in Stage 1, alongside **VLM-based planning and reasoning augmentation** in Stage 2. This design effectively enhances the model’s ability to handle both low-level actions during stage 1 and complex task planning during stage 2. In contrast, GUICourse didn't include inner monologue for trajectory data, which we demonstrate its importance for planning and reasoning in W3.3 and Table 17.
>
> We deeply acknowledge and appreciate the contribution of previous methods such as GUICourse. But we also think this approach differs fundamentally from our work, which focuses on introducing explicit planning and reasoning with unified pure vision design to build a generalizable GUI agent. We truly believe our novel contribution can complement previous efforts  to further advance agent capabilities in real-world evaluations.

---

> ### Author Response · Authors · 2024-11-29
> **Follow Up Response (4/5)**
>
> > **Q3: My primary concern remains with the focus of this work. While it emphasizes large-scale pretraining and various engineering optimizations, it still lacks significant technical innovation regarding the vision perception or language agents.**
>
>
>
> A: Thanks you for your recognition for our training pipeline and engineering optimization effort! Exploring the whole pipeline to build AGUVIS really challengging but exciting. We appreciate your focus on the techinical innovation and we are more than happy to clarify the key contributions of our work and its value to the field!
>
> **The core focus of our work is the development of an autonomous pure vision GUI agent framework capable of generalizing across diverse GUI environments and platforms.** Autonomous means that the agent model needs to generate repliable plan and reason on its own, while pure vision perception with unified coordinate-based action space significantly enhances generalizability across various environments. This is akin to human interaction with GUIs, where actions are based on visual input rather than abstract representations like HTML elements or code snippets. (We comprehensively elaborated these challenges for GUI agents in L51-96.)
>
> **However, one of the major challenges we address is the lack of generalizable agent data due to the fragmented nature of existing GUI agent trajectories.** These trajectories are often environment-specific, recorded with different observation and action formats, making it difficult to use them for training an autonomous agent. For instance, training a model to read HTML or select a target element using its ID is fundamentally different from training a model to click a target on a screen using pixel coordinates. Simply mixing these data formats compromises the model’s ability to generalize across varied modalities.
>
> Furthermore, most prior methods do not integrate explicit planning and reasoning capabilities, often relying on closed-source models or training agents to make decisions without any explicit reasoning process. This dichotomy leads to either weak grounding in GUI interactions or insufficient reasoning abilities. For an **autonomous** agent to function effectively across multiple platforms, incorporating planning and reasoning is essential. Unfortunately, existing trajectory data typically consists only of actions and observations recorded by human annotators, without capturing the explicit reasoning that drives each decision. Annotating such reasoning is costly and labor-intensive, which is why no existing model autonomously completes tasks across different platforms without relying on external reasoning systems. Therefore, currently no agent model can autonomously complete tasks across various real-world platform environments or online evaluations, such as AndroidWorld (Mobile), Mind2Web-Live (Web), and OSWorld (Desktop), without assistance from closed-source models.
>
> **AGUVIS is the first open-source model that addresses these challenges.** It is capable of autonomously completing tasks across diverse GUI environments, including mobile, web, and desktop platforms, with a unified data pipeline and vision-based perception system. Our results on the OSWorld benchmark—evaluating performance in a challenging real desktop environment—demonstrate AGUVIS’s superior performance (10.46), which approaches that of the recently released Claude computer-use API (14.90).
>
> However, we still recognize that this is just the beginning. **Our work on AGUVIS is the first step toward building autonomous GUI agents via imitation learning with large-scale, unified data collection.** We hope that our transparent framework and pipeline will serve as a foundation for future research and inspire further advancements in the field!
>
> ------
>
> > **Q4: Could you provide details on the training costs for 7B/72B models at each stage? Are these models fully trained, or do they utilize LoRA for fine-tuning? Additionally, how does the performance compare across these methods? which will provide valuable reference for community. Thank you.**
>
> A: Thank you so much for your attention to the training cost! All models mentioned in our paper were fully trained, and we shared our training details with cost in Appendix C.2. Since we built AGUVIS as a foundational GUI agent model, we did not use LoRA, as our experience suggests that full fine-tuning generally yields better results. Although we have made every effort to provide as many additional results as possible during discussion phrase, it is challenging to re-implement and train the LoRA baseline for comparison in the limited time remaining. We will strive to include these results in the next version. We sincerely hope for your understanding.
>
> We deeply agree that training cost is challenging for many researchers. We are committed to open-sourcing all our models as foundation models so that others can fine-tune them to specialize them with new abilities and achieve better results.

---

> > ### Author Response · Authors · 2024-11-29
> > **Follow Up Response (5/5)**
> >
> > We would like to sincerely thank you for your thoughtful and detailed feedback. We truly appreciate the time and attention you’ve devoted to reviewing our work and response. Your kind words, such as "I understand the effort it must have taken to provide such comprehensive insights and conduct additional experiments within a short timeframe," are deeply encouraging and motivating. We also greatly appreciate your commitment as a responsible reviewer!
> >
> > Thank you so much for helping us improve our work and discuss with us! We hope our response can address all your concerns. We are more than happy to address any further questions you may have!

---

> > > ### Author Response · Authors · 2024-12-01
> > > **Thank you for your time and effort in reviewing our work and posing insightful questions!**
> > >
> > > Dear Reviewer Vo6b,
> > >
> > > Thank you for your time and effort in reviewing our work and posing insightful questions! We greatly appreciate your engagement in the discussion. As the discussion phase concludes in two days, we would like to ensure our recent responses have addressed all your follow-up questions regarding token consumption, data pipeline explanation, technical innovation, and training costs. We sincerely appreciate your valuable and encouraging feedback during our discussion. We are more than happy to discuss any further questions you may have!
> > >
> > > Authors

---

> > > > ### Author Response · Authors · 2024-12-03
> > > >
> > > > Dear Reviewer Vo6b,
> > > >
> > > > As the discussion phase nears its conclusion **in a few hours**, we would like to take this opportunity to express our sincere gratitude for your valuable feedback and constructive engagement throughout this process.
> > > >
> > > > With only a few hours remaining, we would like to ensure that all your concerns have been thoroughly addressed. **In our previous response, we made a dedicated effort to comprehensively address all follow-up questions, including those regarding token consumption, the data pipeline, technical innovations, and training costs.** If you have the chance, we would greatly appreciate it if you could review our previous response to confirm whether it fully addresses your concerns.
> > > >
> > > > We deeply appreciate the time and thoughtfulness you have invested in this discussion and are eager to hear any further thoughts or suggestions you may have. Thank you once again for your support and attention to our work!
> > > >
> > > > Best regards,
> > > >
> > > > The Authors

---

### Official Review · Reviewer_RjsL · 2024-11-03

**Soundness:** 4
**Presentation:** 3
**Contribution:** 3
**Rating:** 6
**Confidence:** 5

**Summary:**

This paper introduces AGUVIS, a unified, vision-based framework for autonomous GUI agents across multiple platforms. It begins by organizing existing GUI-related datasets and applying carefully designed data augmentations, especially for incorporating low-level reasoning into the existing datasets. The authors then train Qwen2-VL on these organized datasets in two stages: grounding and planning. Experiments conducted on various datasets, including, screenspot, mind2web, etc.,  demonstrate promising results.

**Strengths:**

- The motivation is reasonable. The model should be able to reason before performing grounding and GUI automation tasks. It is also a good idea to incorporate such data in the training stage.
- The organized datasets are valuable. The authors collect most of the existing datasets and the augmentation strategy is interesting.
- The performance is quite promising across several benchmarks, even surpassing very recent papers just released with a large margin.

**Weaknesses:**

1. The formalization presented by the authors is generally good. However, there is one point that needs clarification.
 **Line 123:** The update of "its belief state b_t" is mentioned, but it's unclear which specific part of the model this refers to. This needs further elaboration as it can be confusing.

**Major Concerns.** The primary contribution of this paper is the introduction of the AUGVIS collection which is then used to train Qwen2-VL. However, several crucial details are missing:

2. The authors mention using a VLM to generate the inner monologue for each step in the trajectory. However, it is unclear **which VLM** was used. Current VLMs, including GPT-4o, tend to perform poorly in understanding screenshots, as evidenced by their results in later experiments. **How was the data quality ensured to be reliable?** What is the approximate **accuracy**?

3. The paper mentions that the prompt includes various elements. However, it is unclear how the information is organized and it would be better to just show some **the specific prompt used**.

4. What does the **Planning & Reasoning Trajectories** look like? The authors highlight it in the paper, but I didn't find any examples, so it's difficult to determine how it differs from existing ones. The authors should visualize some samples.

Experiments:

5. The setting of "self-plan" in the ScreenSpot mentioned in the text is somewhat unclear. The author notes that models are required to generate plans based on the original instructions. Specifically, how is this done, and what is the prompt given to the planner? From the results, it is evident that simply adding this mechanism leads to a significant performance improvement, which is excellent. However, the author also needs to analyze the reasons behind this improvement.

6. From Table 6, it seems that the results from Stage 1 training do not significantly impact Mind2Web, as the results for (a) AGUVIS-G and (b) Qwen2-VL are quite similar. However, in Table 1, we can see a noticeable improvement in AGUVIS-G's results in ScreenSpot after Stage 1 training. Could the authors explain the reason for this? Could it be related to the benchmark settings?

7. This work is based on a stronger backbone, Qwen2-VL, compared to others. The author should more clearly highlight the performance improvements contributed by this work. For instance, low-level instructions are a significant contribution. In addition to testing Qwen2-VL (zero-shot) and the final model on ScreenSpot, could a variant be trained without low-level instructions?

8. The authors should add more training details about the proposed model over Qwen2-VL, for example, which modules are frozen, GPU hours, etc.

**Questions:**

1. What is the difference between "Qwen2-VL" and "AGUVIS-G-7B" in Table 1? The former is zero-shot, correct?
2. I notice that sometimes it’s written as AGUVIS and other times as AUGVIS. Is this intentional, or is it a typo?
3. What is the backbone model of "Choice" Gounder? It should be driven by an LLM, correct?
4. In Table 6, AGUVIS-7B's performance is slightly different from that in Table 2. Is this due to some test setting issues?
5. Is AGUVIS-7B compatible with other LLMs like GPT-4? I see different grounding models with LLM; can AGUVIS-7B also achieve this?

---

> ### Author Response · Authors · 2024-11-22
> **Official Comment by Authors (1/3)**
>
> We greatly appreciate your recognition of our work! We are pleased to hear that you acknowledge our motivation for creating AGUVIS, showcasing unified pure vision GUI agent design and autonomous reasoning with grounding capabilities through training with our valuable augmented data collection. To construct the
> collection, we built a data pipeline framework to integrate different data sources into a standardized format, and then implement multiple augmentation strategies we introduced in our paper to augment these datasets. We are continuously adding more datasets to this framework and are preparing to publish this repository with processed training data to benefit future research.
>
> Meanwhile, we noticed that you have some constructive questions about our work. We are pleased to explain further below.
>
> ----
> >**W1: Unclear formalization of belief state update  'belief state $b_t$ '**
>
> A: At L122, we stated:
>
> > At each time step t, the agent receives an image observation $o_t$ from the GUI environment, updates its **belief state $b_t$** based on its previous actions and observations, and generates an inner monologue.
>
> Our original intention was to describe how the model performs internal implicit reasoning based on the current situation and then generates an explicit inner monologue. However, since this does not refer to a specific part of the model output, it might cause confusion with the explicit reasoning thought denoted as $h_t$. We have removed this term to avoid confusion and reflected this change in the uploaded revised version. Thank you for your suggestion!
>
>
> ----
> >**W2: Which VLM was used for inner monologue generation and how data quality was ensured?**
>
> - **During our data curation, we use GPT-4o to augment agent trajectory by adding explicit reasoning thought and low-level instruction.** As shown in newly added Appendix Section B.2, we present the input prompt to GPT-4o for generating detailed, low-level instructions. Specifically, the prompt is consist of high-level goal, previous action history, the current ground truth action command, and the current observation image, where the target object is highlighted with a red box. We then ask GPT-4o to complete the reasoning process and respond in JSON format for easy parsing.
> - According to our Table 1, GPT-4o indeed performs poorly in GUI grounding, but we found that **it performs well in understanding and interpreting actions with GUI screenshots, especially when provided with ground truth action commands and highlighted target objects.** We leveraged this to annotate reasoning process of trajectories.
> - **Our qualitative evaluation shows high accuracy of the augmented data.** Through a human study of samples from each datasets, as described in Appendix Section B.3, 86.7% of the augmented data successfully demonstrated intermediate reasoning that aligned with both the ground truth actions and overall goal intention. Only 7.8% of cases were affected by dataset noise, and 5.5% showed misinterpretation of action intention in clean data.
>
>
> ----
> >**W3&4: Showcase and illustrate the prompt organization and resulting planning & reasoning trajectory examples**
>
> Thank you for your suggestion! In Appendix Figure 4, We illustrate the generated trajectories schema. In Appendix Section C.1, we have visualized our trajectory template to showcase the training prompts. Additionally, we highlight how our AGUVIS collection differs from existing data in Appendix Figure 5.
>
> ----
> >**W5: More examples and explanation of the 'self-plan' setting**
>
> A: We appreciate that you noted the improvement of self-plan setting and it definitely deserve further explanation.
> - In Appendix C.1, we present the training templates for Stage1 and Stage2. We use the special token <|recipient|> along with `os` or `all` to control whether the message content is an inner monologue or a pyautogui action command. Thanks to this design, we can use <|recipient|> during the inference phase to control the content generated by the model.
> - In the self-plan setting, we do not add any word after <|recipient|>, so the model can choose to generate `os` to directly produce a pyautogui command, or generate `all` to first create natural language reasoning and then generate a pyautogui command.
> - To demonstrate the effectiveness of planning, we visualized several hard examples of ScreenSpot in Appendix E.2.2. These examples illustrate how an additional reasoning step helps the model achieve more accurate grounding.

---

> > ### Author Response · Authors · 2024-11-22
> > **Official Comment by Authors (2/3)**
> >
> > ----
> > > **W6: Stage 1 does not significantly impact the performance on Mind2Web in Table 6 but does on ScreenSpot in Table 1.**
> >
> > A: Thanks for your insightful feedback! This phenomenon is because:
> > - ScreenSpot, as a benchmark designed for comprehensive GUI grounding evaluation, it benefits significantly from Stage 1 training, which enhances grounding abilities. This targeted improvement leads to a substantial performance boost on ScreenSpot.
> > - In contrast, Mind2Web emphasizes planning over grounding and operates at a fixed resolution in the web domain. Since Stage 1 focuses on grounding rather than planning, its impact on Mind2Web is limited, with significant improvements only appearing after Stage 2.
> >
> >
> >
> > ----
> >
> > > **W7: The author should more clearly highlight the performance improvements contributed by Qwen2-VL. For instance, could a variant be trained without low-level instructions?**
> >
> > A: Thank you for your insightful feedback! We agree that it's important to highlight the performance improvements contributed by Qwen2-VL. We demonstrated in Section 4.2 that our method is model-agnostic by using a weaker LLaVA backbone.
> >
> > - Furthermore, a variant be trained without inner monologue (including low-level instruction) can also be a method to analyze performance improvement. To address your suggestion of analyzing the impact of low-level instructions, we also updated ablation studies by training a variant of our model without the inner monologue (which includes low-level instructions). We believe that the inner monologue plays a crucial role due to two main factors:
> >     1. The generation of an inner monologue elicits the reasoning process concerning current actions.
> >     2. The produced reasoning inner monologue provides informative context that informs subsequent planning stages.
> >
> >     To validate the importance of the inner monologue and low-level instructions, we present the following ablation results:
> >
> > | Setting | ScreenSpot | Multimodal-Mind2Web ||| AndroidControl ||
> > |---|---|---|---|---|---|---|
> > | | | Cross-Task | Cross-Website | Cross-Domain | High-Level | Low-Level |
> > |AGUVIS|84.4|58.5|55.4|54.8|61.5|80.5|
> > |AGUVIS w/o IM|79.3|55.4|53.7|54.9|60.3|69.1|
> >
> >
> > Our findings indicate that:
> >     - Inner monologue clearly enhances performance on planning benchmarks.
> >     - Inner monologue can also improves low-level instruction grounding. This is because the low-level instructions in our augmented trajectory act as atomic instruction and grounding action pairs, enhancing GUI grounding ability.
> >
> > Overall, this ablation study clearly demonstrates the performance improvements brought by incorporating the inner monologue and low-level instructions in training, highlighting the contributions of Qwen-2 VL.
> >
> > Additionally, we recognize the importance of demonstrating that our method is model-agnostic. In Appendix E.1.1, we further provide a comprehensive ablation studies using the LLaVA backbone. These studies show that the AGUVIS pipeline can effectively build a comparable GUI agent model using a weaker backbone, further justifying the versatility of our approach.
> >
> > ----
> > > **W8: More details of training.**
> >
> > A:
> > - Following the SFT strategy in Qwen2-VL, we freeze the ViT parameters during training. We train AGUVIS on a cluster of H100-80G GPUs: AGUVIS-7B uses 8 nodes and completes the grounding training within 5 hours and planning \& reasoning training within 1 hour.
> > AGUVIS-72B uses 16 nodes and completes the grounding training within 30 hours and planning \& reasoning training within 6 hours.
> > - We have added more training details in Appendix C.2.
> >
> >
> > ----
> > > **Q1: What's the difference between "Qwen2-VL" and "AGUVIS-G-7B" in Table 1?**
> >
> > A: Yes, Qwen2-VL represents the zero-shot GUI grounding performance on ScreenSpot. In contrast, AGUVIS-G-7B has completed stage 1 training.
> >
> > ---
> > > **Q2: Is AGUVIS vs AUGVIS intentional or a typo?**
> >
> > A: Thank you for catching that typo! We've corrected it in the revised version.
> >
> > ---
> > > **Q3: What is the backbone model of "Choice" Grounder?**
> >
> > A: The results are derived from SeeAct[1], which uses a DeBERTa-base cross-encoder to rank the interactable elements on the current HTML page. It selects the top 50 elements as choices, and the GPT-4(o) model then picks one of these elements as the answer. We have added citations for each method in Table 2 to enhance its readability.
> >
> > References:
> > [1] GPT-4V(ision) is a Generalist Web Agent, if Grounded. Zheng et al., 2024.

---

> > > ### Author Response · Authors · 2024-11-22
> > > **Official Comment by Authors (3/3)**
> > >
> > > ---
> > > > **Q4: Why are AGUVIS-7B's performance numbers different between Tables 2 and 6?**
> > >
> > > A: Since Table 2 involves evaluating the larger AGUVIS 72B model, directly using the huggingface generate API is very costly. Therefore, we used the vLLM framework to evaluate both the 7B and 72B models in Table 2 for consistent setting in same table. However, Table 6 only involves the evaluation of the 7B model, so we used the hf generate API. The design differences between vLLM and hf generate lead to slight fluctuations in the results. However, we strictly ensure consistent settings within the same table. The fluctuations between the two tables do not affect our conclusions.
> > >
> > > ---
> > > > **Q5: Is AGUVIS-7B compatible with other LLMs like GPT-4?**
> > >
> > > A: Yes! We demonstrated this in Tables 4 and 5 for both web and Android online evaluations. In these evaluations, we also used the GPT-4o model to plan low-level instructions, while AGUVIS followed and generated the action commands. This illustrates AGUVIS's compatibility, showing that it can function not only as an independent autonomous GUI agent but also as a grounding model. This versatility is due to its flexible data schema design, as detailed in Appendix Section D.3.1.
> > >
> > >
> > > ----
> > >
> > > We sincerely appreciate your detailed feedback. We hope the above response can address all your concerns. If you have any questions, we are pleased to provide further clarification!

---

> ### Author Response · Authors · 2024-11-25
>
> Dear Reviewer RjsL:
>
> We sincerely thank you for these constructive comments and evaluation of our paper. With the ICLR public discussion phase ending in **two days**, we kindly ask you to take a look at our responses. Our rebuttal provided more clarification of our framework and additional experiments in response to your concerns. Please let us know whether our response addresses your concerns or whether there is any further detail we can provide to help address these concerns.
>
> Thank you again for dedicating your time to reviewing our paper.

---

> > ### Comment · Reviewer_RjsL · 2024-11-26
> >
> > Thank you to the authors for the detailed rebuttal and the expanded appendix, which addressed most of my concerns. I have one follow-up question:
> >
> > The results of W7 are quite interesting. Adding annotations for low-level instructions seems to significantly improve grounding or low-level planning rather than high-level grounding. Do the authors have any thoughts on why the improvement on Mind2Web is relatively limited?

---

> > > ### Author Response · Authors · 2024-11-26
> > >
> > > We are sincerely happy to hear that our response can address most concerns! We are also pleased to answer this insightful follow-up question.
> > >
> > > > The results of W7 are quite interesting. Adding annotations for low-level instructions seems to significantly improve grounding or low-level planning rather than high-level grounding. Do the authors have any thoughts on why the improvement on Mind2Web is relatively limited?
> > >
> > >
> > > Thank you for your thoughtful observation! We find these results insightful as well, and we would like to explain this phenomenon from two perspectives: **benchmark characteristics** and **the source of abilities during training**.
> > >
> > > ---
> > >
> > > ### 1. **Benchmark Characteristics**
> > > We designed this ablation study using **ScreenSpot**, **Mind2Web**, and **AndroidControl** because these benchmarks reflect different aspects of agent abilities:
> > >
> > > - **ScreenSpot**:
> > >   This benchmark evaluates **non-contextual single-step grounding**, represented as $(o, a^{inst}, a)$, where the observation ($o$), low-level instruction ($a^{inst}$), and PyAutoGUI action ($a$) form a grounding tuple. It primarily focuses on direct grounding capabilities without multi-step context.
> > >
> > > - **AndroidControl**:
> > >   This benchmark uses **multi-step trajectories** and evaluates in two modes:
> > >   - **High-Level Mode**: The agent is given high-level goals ($G$) and the current observation ($o_t$), e.g., $[G, \dots, o_t]$, to predict the next PyAutoGUI action ($a_t$).
> > >   - **Low-Level Mode**: In addition to high-level goals and observations, the agent receives ground truth low-level instructions ($a_t^{inst}$), e.g., $[G, \dots, o_t, a_t^{inst}]$, to predict $a_t$. This mode is more akin to **contextual grounding** compared to the non-contextual grounding of ScreenSpot.
> > >
> > > - **Mind2Web**:
> > >   This benchmark resembles the **high-level mode of AndroidControl**, where the agent must infer the next action based on its understanding of the high-level goal and the current state without any intermediate instructions. This task requires **significant planning and reasoning** capabilities since Mind2Web includes more long-term hard tasks.
> > >
> > > These benchmarks highlight different aspects of agent performance, allowing us to observe how low-level instruction augmentation influences grounding and planning across diverse scenarios.
> > >
> > > ---
> > >
> > > ### 2. **Source of Abilities During Training**
> > >
> > > The **inner monologue augmentation** applied to stage 2 trajectory data transforms the original sequences:
> > >
> > > > $[G, o_1, a_1, o_2, a_2, \dots]$
> > >
> > > Into augmented sequences:
> > >
> > > > $[G, o_1, a_1^{inst}, a_1, o_2, a_2^{inst}, a_2, \dots]$
> > >
> > > This augmented format turns the trajectory into a sequence of **contextual grounding pairs** embedded within the framework of achieving a high-level goal. Each pair ($o_t, a_t^{inst}, a_t$) establishes a clear mapping between the observation, low-level instruction, and action. This structure directly enhances the model's ability to perform **low-level instruction grounding**, which is reflected in:
> > >
> > > - **Non-Contextual Grounding**: Improvements on ScreenSpot, as the grounding pairs help establish a direct and explicit connection between low-level instructions and actions.
> > > - **Contextual Low-Level Instruction Grounding**: Significant improvements in the low-level mode of AndroidControl, which benefits from the contextual nature of the augmented trajectories.
> > >
> > > For high-level modes such as AndroidControl High-Level and Mind2Web, the improvements from inner monologue augmentation come from two additional factors:
> > > 1. **Eliciting Reasoning Ability**: The explicit low-level instructions enhance the agent's reasoning capabilities, enabling it to better decompose complex tasks.
> > > 2. **Informative Action History**: The augmented low-level instructions act as a detailed and structured action history, providing the model with richer context to plan the next steps.
> > >
> > > These two factors lead to measurable improvements in planning-heavy tasks like AndroidControl High-Level and Mind2Web.
> > >
> > > We believe these insights underscores the significant role of inner monologue augmentation in boosting both high-level and low-level performance. It also opens avenues for further exploration on GUI agent research:
> > > - **Exploiting Inner Monologue**: We see potential in developing advanced training strategies to fully leverage AGUVIS’s inner monologue, drawing inspiration from recent research on reasoning and planning in math and code.
> > > - **Explainability**: The explicit inner monologue improves the GUI agent’s explainability, allowing for better analysis, improvement, and oversight of its behavior, rather than solely relying on pyautogui command actions.
> > >
> > > These findings reaffirm the value of this design in advancing both the planning and reasoning capabilities of AGUVIS, paving the way for more robust and generalizable GUI agents.
> > >
> > > Thank you again for raising this excellent question!

---

> > > > ### Author Response · Authors · 2024-12-02
> > > >
> > > > Dear Reviewer RjsL
> > > >
> > > > We sincerely appreciate your time and effort in reviewing our paper! With the ICLR public discussion phase ending **in one day**, we would like to ensure our previous response addressed your questions regarding analysis of performance improvement. If you have any further questions or additional feedback today, we would be more than happy to address them. Thank you!
> > > >
> > > > Authors

---

### Official Review · Reviewer_KN7M · 2024-11-03

**Soundness:** 3
**Presentation:** 3
**Contribution:** 2
**Rating:** 5
**Confidence:** 4

**Summary:**

AGUVIS is a novel framework for building autonomous agents that interact with Graphical User Interfaces (GUIs) using a purely visual approach. AGUVIS utilizes image-based observations and grounding and by employing a unified action space with a plugin system, AGUVIS can operate across various platforms.

**Strengths:**

1. Extensive evaluations on different benchmarks;
2. Promising performances on benchmarks;
3. Interesting to see the usage of pyautogui as the bridge to unify the action space.
4. Promised to open-source (not yet)

**Weaknesses:**

1. overall, I did not find much difference compared to previous methods like AMEX, SeeClick, etc. They also use the pure-vision representation and use the both grounding and planning dataset to train the GUI agents.
2. The authors proposed to use the VLM to construct the CoT dataset, but how are the gains? Will the CoT training will additional performance gain?
3. The author chose the pyautogui to unifiy the action space, but the experiments did not give the readers some examples to showcase its advantages, something like how this could help to adapt to some new actions?

**Questions:**

1. In table Six, how about only training with stage 2?
2. How can AGUVIS be applied to real-world scenarios beyond the evaluated benchmarks? Like closing the ads
3. I am very curious that what's the reason behind the model size choice, like in Table 4 and Table 5, Grounder is 7B, while when e2e it is 70B?
4. When deployed for Phone and Web, should we have some specific designs respectively?

---

> ### Author Response · Authors · 2024-11-22
> **Official Comment by Authors (1/3)**
>
> Thank you for recognizing our work and providing constructive feedback! We're delighted to hear about your interest in our unification design and your acknowledgment of our extensive evaluation efforts to verify the model's promising performance. Creating a unified model is a significant challenge, as it requires unifying both the model's training data and the evaluation framework. We developed a comprehensive offline GUI agent evaluation framework and integrated our pure vision model into online evaluation environments for each platform. These, along with our training data, code, and models, should serve as valuable contributions to the community. We are in the process of releasing all these resources to support further research in general autonomous GUI agent studies.
>
> We also noticed you have some constructive questions about our work, and we're happy to elaborate further below!
>
> ---
> > **W1: Comparison to previous methods like AMEX and SeeClick which also use pure-vision representation and both grounding/planning datasets**
>
>
> A: Both of these previous work laid important groundwork in vision-based GUI interaction. Compared with these work, we would like to highlight two foundamental aspects of our work:
> - **Thanks to the unified design, AGUVIS is a general cross-platform pure-vision based agent instead of specialized agent for single platform for either desktop or mobile.** Through unified training with pluggable action space, our model can be directly deployed on these platform. In contrast, AMEX is designed for mobile and SeeClick is required to fine tune for each environment/benchmark.
> - **AGUVIS introduces a novel explicit planning and reasoning framework that differs from previous approaches.**  We illustrate these differences with visual examples in Figure 5. While earlier approaches fine-tune models with multi-step trajectories, they typically generate action commands directly, bypassing the reasoning process and low-level instructions. On the one hand, the previous approach fails to elicit the reasoning ability of VLMs during generation; On the other hand, this force the model to predict next action conditioning on less informative coordinates action representation(e.g. `pyautogui.click(0.868, 0.610)`) rather than informative natural language low-level instruction (e.g. `click on the buy button.`). Therefore, we propose introducing an explicit reasoning process with a large-scale data collection to improve pure vision GUI agents by elicit reasoning ability and also use the low-level instruction as informative previous action history. Moreover, this explicit reasoning process provides explainability for analyzing, improving, and overseeing GUI agent behaviors. We believe this approach could serve as a foundation for future GUI agent research.
>
> ----
>
> > **W2: Unclear benefits of using VLM to construct CoT dataset**
>
> A: Thank you for your insightful feedback. We think that the role of inner monologue can be attributed to two factors:
> - The generation of an inner monologue elicits the reasoning process concerning current actions.
> - The produced reasoning inner monologue provides informative context that informs subsequent planning stages.
>
> To further validate the importance of inner monologue, we conducted ablation studies to demonstrate the performance improvements achieved by incorporating inner monologue (IM) in training:
>
> | Setting | ScreenSpot | Multimodal-Mind2Web ||| AndroidControl ||
> |---|---|---|---|---|---|---|
> | | | Cross-Task | Cross-Website | Cross-Domain | High-Level | Low-Level |
> |AGUVIS|84.4|58.5|55.4|54.8|61.5|80.5|
> |AGUVIS w/o IM|79.3|55.4|53.7|54.9|60.3|69.1|
>
> Our findings indicate that:
> - Inner monologue clearly enhances performance on planning benchmarks.
> - Inner monologue can also improves low-level instruction grounding. This is because the low-level instructions in our augmented trajectory act as atomic instruction and grounding action pairs, enhancing GUI grounding ability of model.
>
> Overall, this ablation study clearly demonstrates the performance improvements brought by incorporating inner monologue in training data, further justifying its effectiveness and supporting our analysis.
>
> ----
>
>
> > **W3: Lack of examples showcasing pyautogui's advantages for adaptation to new actions.**
>
> A: PyAutoGUI has predefined GUI-related atomic operations, which can be effectively transferred to new GUI interfaces. When applied to new environments like mobile, additional actions can be seamlessly integrated into the system prompt in a plugin format, allowing the model to utilize new actions. This approach enables the agent model to combine internally learned PyAutoGUI actions with newly added ones to complete tasks efficiently. We have included the mobile environment prompt used for Android World in Appendix Section A.2 as an example of adapting to new actions.
>
> ---

---

> > ### Author Response · Authors · 2024-11-22
> > **Official Comment by Authors (2/3)**
> >
> > ---
> >
> > > **Q1: In Table 6(Table 7 in revised version), how about only training with stage 2?**
> >
> > A: Based on Table 6 (Table 7 in revised version), we further added experiments that only train stage 2 (AGUVIS w/o Stage 1), with results shown in the table below.
> >
> > | Setting | ScreenSpot | Multimodal-Mind2Web ||| AndroidControl ||
> > |---|---|---|---|---|---|---|
> > | | | Cross-Task | Cross-Website | Cross-Domain | High-Level | Low-Level |
> > |AGUVIS|84.4|58.5|55.4|54.8|61.5|80.5|
> > |AGUVIS w/o Stage 1|77.4|59.7|55.3|55.8|58.8|79.8|
> > |AGUVIS w/o Stage 2|81.8|50.9|45.2|45.3|58.0|75.6|
> >
> > Analysis of Results:
> > 1. Impact on ScreenSpot (GUI Grounding Performance):
> >     - The performance on the ScreenSpot benchmark drops from 84.4% to 77.4% when Stage 1 is omitted.
> >     - ScreenSpot evaluates GUI grounding across a diverse set of images and domains, including web, desktop, and mobile interfaces. Stage 1 provides extensive and diverse GUI grounding training data, which is crucial for high performance on this benchmark. Without Stage 1, the model lacks the necessary exposure to varied GUI elements, leading to reduced grounding capabilities.
> > 2. Impact on MM-Mind2Web and AndroidControl (Planning and Reasoning Tasks):
> >     - Performance maintained without stage 1.
> >     - Potiential two reasons:
> >         - Potiential Pre-trained capabilities of Qwen2-VL: The backbone model Qwen2-VL was pre-trained on natural image grounding tasks, which provides it with inherent grounding abilities even without Stage 1 GUI grounding training.
> >         - Extensive Stage 2 Trajectory Data: Stage 2 involves a large amount of trajectory data including grounding action pairs. This extensive training enables the model to effectively handle grounding in Mind2Web and AndroidControl, even in the absence of Stage 1.
> >
> > To further support our analysis, we refer to the ablation study conducted on the LLaVA model (as presented in Table 8 of our paper). The results are as follows:
> >
> > | Setting | ScreenSpot | Multimodal-Mind2Web |||
> > |---|---|---|---|---|
> > | | | Cross-Task | Cross-Website | Cross-Domain |
> > |**AGUVIS-LLaVA**|81.2|55.3|50|50.8|
> > |**AGUVIS-LLaVA w/o Stage 1**|71.3|42.5|40.3|42.8|
> > |**AGUVIS-LLaVA w/o Stage 2**|70|43.4|39.0|40.7|
> >
> > We found that LLaVA model shows significant performance drops on both ScreenSpot and MM-Mind2Web when either Stage 1 or Stage 2 is omitted. Based on these results with two backbones, we can conclude that:
> > - Qwen2-VL: Due to its pre-training on natural image grounding, it can maintain reasonable performance on planning tasks without Stage 1.
> > - LLaVA: Lacks such pre-training, thus both Stage 1 and Stage 2 are critical for achieving high performance.
> >
> >
> > Additionally, the consistent improvements across both Qwen2-VL and LLaVA backbones demonstrate that our AGUVIS methodology is effective regardless of the underlying model, which highlights the universality and adaptability of our training approach.
> >
> > These findings underscore the importance of incorporating both Stage 1 and Stage 2 in the training pipeline to achieve optimal performance across diverse GUI tasks and benchmarks. By providing comprehensive GUI grounding and enhancing planning and reasoning abilities, our AGUVIS approach ensures that models can generalize effectively to various domains and platforms.

---

> > > ### Author Response · Authors · 2024-11-22
> > > **Official Comment by Authors (3/3)**
> > >
> > > ---
> > >
> > > > **Q2: How can AGUVIS be applied to real-world scenarios (e.g. closing ads)?**
> > >
> > > A: Thank you for your suggestion on applying AGUVIS to real-world scenarios. Implementing AGUVIS in practical applications is indeed our primary goal.
> > >
> > > - We would like to first highlight that **the online evaluations in Tables 4 and 5 reflect real-world performance** because AGUVIS interacts with realistic web pages or Android devices to complete GUI tasks in these settings. We have visualized this process in Appendix Section E.3. While we haven't specifically used AGUVIS to close ads, we have encountered similar tasks during online evaluation, such as closing cookie pop-ups, which are out-of-domain situations. This example, which we visualized in Appendix Section E.4, demonstrates the effectiveness of our model in real-world scenarios.
> > > - Moreover, we applied AGUVIS in the real-world benchmark OSWorld, an online evaluation within a realistic operating system environment. As demonstrated in the newly added Table 6 OSWorld, the performance is very promising for both the GPT-4o planner with the AGUVIS-7B grounder and the end2end AGUVIS-72B. These results make AGUVIS-72B the only open-source, pure vision GUI agent capable of completing tasks without GPT-4o planning, achieving performance levels close to the recently released Claude-3-5-Sonnet computer use API.
> > >
> > > | Planner         | Grounding      | Task SR |
> > > |-----------------|----------------|---------|
> > > | GPT-4o          | SoM            | 4.59    |
> > > | GPT-4o          | N/A          | 5.03    |
> > > | GPT-4V          | N/A         | 5.26    |
> > > | Gemini-Pro-1.5  | N/A         | 5.40    |
> > > | GPT-4o          | **AGUVIS-7B**      | 11.07   |
> > > | **AGUVIS-72B**       | N/A     | 10.26   |
> > > | claude-3-5-sonnet-20241022 | N/A     | 14.9   |
> > >
> > >
> > > ---
> > >
> > > > **Q3: Why choose different model sizes in Table 4&5 (7B for Grounder, 70B for e2e)?**
> > >
> > > A: The choice of model size is based on task complexity. GUI grounding with specific low-level instructions is relatively straightforward, so we use a 7B model for this task. As shown in the results of Table 1, AGUVIS-7B is an efficient choice. In contrast, the end-to-end agent process is more complex, requiring the model to decompose high-level goals then plan and determine the next action based on the current situation. Therefore, we use a AGUVIS-72B for the end-to-end realistic online evaluation.
> > >
> > >
> > > ---
> > >
> > > > **Q4: Should there be specific designs for Phone and Web deployment?**
> > >
> > > A: As we mentioned in W3, thanks to pure vision modeling and unified action space, AGUVIS can perform tasks with pyautogui atomic operations in both mobile and web environments. However, compared with web environment, additional action space for mobile environment can indeed help the model better accomplish more complex cross-app tasks, such as navigation keys (mobile.home(), mobile.back(), etc.). Thanks to our pluggable action space design, when applied to environments like mobile, additional actions can be added directly to the system prompt in a plugin format to enable the model to use new actions. We have included the mobile environment prompt used for Android World in Appendix Section A.2 as an example of adapting to new actions.
> > >
> > > ----
> > >
> > > We sincerely appreciate your detailed feedback. We hope the above response can address all your concerns. If you have any questions, we are pleased to provide further clarification!

---

> > > > ### Comment · Reviewer_KN7M · 2024-11-25
> > > > **thanks for the reviewer's response and effort**
> > > >
> > > > I have some follow-up questions:
> > > >
> > > > 1. "PyAutoGUI has predefined GUI-related atomic operations, which can be effectively transferred to new GUI interfaces."
> > > >
> > > > It seems that PyAutoGUI could serve as a unified representation space for different GUI tasks, then we can utilize the data from different data sources. And we should convert the original annotations into the PyAutoGUI format? Am I right?
> > > >
> > > > 2. "Inner monologue can also improves low-level instruction grounding. This is because the low-level instructions in our augmented trajectory act as atomic instruction and grounding action pairs, enhancing GUI grounding ability of model." -> This seems the "Inner monologue" brings additional training data for the grounding task. Did I understand correct?
> > > >
> > > > 3. For my original Q4, another follow-up question is that, the Web and Phone usually also have different image sizes. How the AGUVIS handle this?
> > > >
> > > > 4. Can the author provide some additional information like the average inference time for the task execution?

---

> > > > > ### Author Response · Authors · 2024-11-25
> > > > >
> > > > > Thank you so much for your follow-up response!
> > > > >
> > > > > ---
> > > > >
> > > > > > **1. We can utilize the data from different data sources by converting the original annotations into the PyAutoGUI format**
> > > > >
> > > > > A: Certainly! In our work, we developed a data toolkit designed to systematically unify various sources into a standardized data schema. We then applied template-based grounding augmentation and VLM-based inner monologue augmentation to export trajectories in the PyAutoGUI format.
> > > > >
> > > > > We are committed to open-sourcing this toolkit with AGUVIS collection data, allowing us to actively integrate more future data sources and continuously scale GUI agent trajectories. We hope our AGUVIS collection will be valuable for future GUI agent research.
> > > > >
> > > > > ---
> > > > >
> > > > > > **2. The "Inner monologue" brings additional training data for the grounding task?**
> > > > >
> > > > > A: Yes, in our framework, inner monologue augmentation for stage 2 trajectories data effectively introduces additional training signal for grounding tasks. Let me explain this in detail:
> > > > >
> > > > > Grounding pairs in stage 1 are structured as discrete tuples of **observation** ($o$), **low-level instruction** ($a^{inst}$), and **PyAutoGUI action** ($a$):
> > > > >
> > > > > > $(o, a^{inst}, a)$
> > > > >
> > > > > In most original multi-step agent trajectory data, the focus is on a sequence of **high-level goals** ($G$), observations ($o_t$), and corresponding PyAutoGUI actions ($a_t$), structured as:
> > > > >
> > > > > > $[G, o_1, a_1, o_2, a_2, \dots]$
> > > > >
> > > > > When we augment these trajectories with **inner monologue** for stage 2, which introduces **low-level instructions** ($a_t^{inst}$), the trajectory becomes:
> > > > >
> > > > > > $[G, o_1, a_1^{inst}, a_1, o_2, a_2^{inst}, a_2, \dots]$
> > > > >
> > > > > This augmented structure effectively transforms the trajectory into a sequence of high-quality grounding pairs embedded within **the context of achieving a high-level goal**. Each grounding pair ($o_t, a_t^{inst}, a_t$) provides a clear mapping between the observation, low-level instruction, and the corresponding action, significantly enhancing the model’s low-level instruction grounding capability, which reflected in ScreenSpot benchmark.
> > > > >
> > > > > In summary, the **inner monologue** augmentation does indeed enrich the trajectories with high-quality explicit grounding pairs, making the connection between low-level instructions and actions clearer and more actionable, thereby improving grounding performance.
> > > > >
> > > > >
> > > > > ---
> > > > >
> > > > > > **3. The Web and Phone usually also have different image sizes. How the AGUVIS handle this?**
> > > > >
> > > > >
> > > > > A: The image encoder in AGUVIS is **NaViT** [1], which uses a **dynamic resolution strategy** to encode images. This strategy converts images into a variable number of visual tokens while **preserving their original aspect ratios and sizes** (more details are available in the NaViT paper).
> > > > >
> > > > > Preserving the original aspect ratio and high resolution is crucial for accurate GUI understanding and grounding. NaViT’s dynamic resolution strategy ensures AGUVIS can handle varying image sizes effectively. To balance efficiency and performance, we cap the maximum number of visual tokens at **1,196** (suitable for 720p resolution).
> > > > >
> > > > > Reference
> > > > >
> > > > > [1] Patch n' Pack: NaViT, a Vision Transformer for any Aspect Ratio and Resolution. Dehghani et al., 2024.
> > > > >
> > > > > > 4. Can the author provide some additional information like the average inference time for the task execution?
> > > > >
> > > > > A: We would first like to clarify the components of execution time for agent tasks:
> > > > >
> > > > > - For **online agent tasks**, execution time is primarily composed of two factors:
> > > > >   1. **Agent model inference time**
> > > > >   2. **Environment execution time**
> > > > >
> > > > > - For **offline agent evaluations**, where the agent does not interact with the environment, the execution time depends solely on the **agent model inference time**.
> > > > >
> > > > > To provide realistic performance metrics, we report the **average time cost per step** from our OSWorld online evaluation, conducted in a real Ubuntu OS environment.
> > > > >
> > > > > - **Agent Model Inference Time**:  Our AGUVIS model is served using **vLLM** on 4 A100-80G GPUs, enabling efficient, low-latency inference.
> > > > >
> > > > > - **Environment Execution Time**:  AGUVIS’s pure vision approach relies on screenshots, avoiding the additional time cost associated with extracting accessibility trees, which can be resource-intensive.
> > > > >
> > > > > Under these settings, AGUVIS achieves an **average inference time of approximately 4.89 seconds per step** across 369 Ubuntu tasks in OSWorld. This reflects its capability to handle real-world OS-level tasks efficiently.
> > > > >
> > > > > -----
> > > > >
> > > > > We hope this can address your follow-up concerns, and we are pleased to provide further clarification!

---

> > > > > > ### Author Response · Authors · 2024-12-01
> > > > > > **Thanks for your time and effort in reviewing our paper and discussing valuable follow-up questions!**
> > > > > >
> > > > > > Dear Reviewer KN7M,
> > > > > >
> > > > > > We sincerely appreciate your time and effort in reviewing our paper and discussing valuable follow-up questions! With the ICLR public discussion phase ending in two days, we would like to ensure our previous response addressed your follow-up questions regarding PyAutoGUI as a unified format, inner monologue for grounding, image size encoding strategy, and execution time. If you have any further questions or additional feedback, we would be more than happy to address them!
> > > > > >
> > > > > > Authors

---

> ### Author Response · Authors · 2024-12-03
>
> Dear Reviewer KN7M
>
> As the discussion phase nears its conclusion in **a few hours**, we would like to sincerely thank you for your thoughtful feedback and constructive engagement throughout this process.
>
> With the final hours upon us, this is our last chance to receive and address any remaining concerns or questions you may have. If you have a moment, we kindly invite you to review our previous response and let us know if it fully addresses your feedback.
>
> We deeply appreciate the time and effort you’ve devoted to this discussion and look forward to hearing your thoughts!
>
> Best regards,
>
> The Authors

---

### Author Response · Authors · 2024-11-22
**Updated Manuscript and Response to All Reviewers**

We sincerely thank all the reviewers for their thoughtful and constructive feedback on our work. We are delighted that our efforts to develop AGUVIS, a unified pure vision GUI agent, have been appreciated, especially regarding its novel contributions to planning, grounding, and reasoning through an two-stage training process. We appreciate `RjsL` and `Vo6b` acknowledging our roadmap for building effective pure vision agents with curated data and two-stage training strategies, `RjsL` and `pn3z` for noting the main challenge of planning and reasoning for GUI agents and recognizing our approach in solving this challenge by incorporating augmented data, `RjsL`, `Vo6b`, `pn3z` for providing positive feedback on the extensive evaluation and superior performance, and `KN7M` and `RjsL` for highlighting the contribution of open-source to the community of unified training datasets. We sincerely thank all the reviewers for their insightful comments and constructive feedback.


## Key Appendix Updates Based on Feedback
In response to the reviews, we have addressed all concerns in our manuscript and added comprehensive details and explanations in our Appendix:


1. Enhanced Visualization and Examples:
    - Added detailed visualizations of training schema, online evaluation processes, and planning trajectories in the Appendix.
    - Included qualitative examples to illustrate real-world applications and error case analyses.
2. Qualitative human study for Data Augmentation Pipeline:
    - Evaluated accuracy of our data augmentation pipeline (Appendix B.3) to further confirm the correctness of inner monologue.
3. Extended Experiments and Ablations:
    - Conducted additional ablation studies to explore the impact of inner monologues, two-stage training, and dataset contributions from different platforms.
    - Model-Agnostic Demonstration: Further validated AGUVIS's methodology on multiple backbone VLMs (e.g., LLaVA and Qwen2-VL), demonstrating its generalizability.

*For clarity, updates in the revised version are highlighted in blue.*

We believe the revised manuscript addresses all concerns raised by the reviewers and highlights AGUVIS's strengths as a comprehensive roadmap for the development of autonomous pure vision GUI agents. We remain committed to open-sourcing the training pipeline, data, and models, ensuring that our contributions benefit the broader research community.

Moreover, we are pleased to report that AGUVIS-72B demonstrates promising performance in the real-world benchmark **OSWorld**, an online evaluation within a realistic operating system environment. **As an end-to-end pure vision GUI agent, AGUVIS-72B achieves performance levels of 10.26%, approaching the recently released Claude-3-5-Sonnet computer using API, making it the only open-source agent capable of completing tasks without relying on GPT-4o planning.** We have updated these results as Table 6 OSWorld. These encouraging results motivate us to continue advancing research in this field.

Thank you for your insightful feedback to help us enrich our work. We hope the revised submission meets your expectations and demonstrates AGUVIS's potential as a valuable foundation for future research in autonomous GUI agent modeling.

---

### Meta-Review · Area_Chair_EyUZ · 2024-12-19

**Metareview:**

The paper presents Aguvis, a unified pure vision-based framework for autonomous GUI agents. The reviewers raise several concerns, and while the authors provide rebuttals and additional explanations, some key issues remain unresolved.

1. **Novelty**: The reviewers question the novelty of the work, suggesting that it mainly involves pre-training VLMs with data combinations. Although the authors highlight the novel integration pipeline and pure vision agent model, the lack of significant technical innovation in vision perception or language agents is a concern.

2. **Data curation and training details**: The paper lacks sufficient details on data curation, such as the accuracy of the VLM-generated inner monologue and the training schedule for each stage. The authors provide some clarifications, but the overall lack of detail may limit the reproducibility and understanding of the work.

3. **Experimental analysis**: The reviewers request further analysis, such as the justification for the two-stage training paradigm and the effectiveness of the inner monologue. The authors conduct additional experiments, but the results do not fully address the concerns, and some explanations are not entirely convincing.

The paper has some strengths, including good performance and a clear writing style. However, the concerns regarding novelty, data curation, and experimental analysis are significant. While the authors have made efforts to address the reviewers' comments, the overall quality of the paper does not meet the standards for acceptance at this time.

**Additional Comments On Reviewer Discussion:**

Reviewers raised concerns about the paper's novelty, data curation, experimental analysis, and training details. They questioned the lack of significant technical innovation and the adequacy of the two-stage training paradigm. Authors responded with additional experiments, explanations, and appendix content. They detailed the use of GPT-4o for inner monologue generation, clarified training schedules, and provided more on token consumption. However, some responses like the justification for two-stage training and the overall novelty contribution remained somewhat unconvincing.

---

### Decision · Program_Chairs · 2025-01-22

Reject